# Development of a natural product optimization strategy for inhibitors against MraY, a promising antibacterial target

Kazuki Yamamoto [1,2] ✉, Toyotaka Sato[3,4,5], Aili Hao [6], Kenta Asao[1], Rintaro Kaguchi [1], Shintaro Kusaka[1], Radhakrishnam Raju Ruddarraju[1], Daichi Kazamori[7], Kiki Seo[7], Satoshi Takahashi[8,9], Motohiro Horiuchi[3,4,5], Shin-ichi Yokota [10], Seok-Yong Lee [6] & Satoshi Ichikawa [1,2,11] ✉

MraY (phospho-*N*-acetylmuramoyl-pentapeptide-transferase) inhibitory natural products are attractive molecules as candidates for a new class of antibacterial agents to combat antimicrobial-resistant bacteria. Structural optimization of these natural products is required to improve their drug-like properties for therapeutic use. However, chemical modifications of these natural products are painstaking tasks due to complex synthetic processes, which is a bottleneck in advancing natural products to the clinic. Here, we develop a strategy for a comprehensive in situ evaluation of the build-up library, which enables us to streamline the preparation of the analogue library and directly assess its biological activities. We apply this approach to a series of MraY inhibitory natural products. Through construction and evaluation of the 686-compound library, we identify promising analogues that exhibit potent and broad-spectrum antibacterial activity against highly drug-resistant strains in vitro as well as in vivo in an acute thigh infection model. Structures of the MraY-analogue complexes reveal distinct interaction patterns, suggesting that these analogues represent MraY inhibitors with unique binding modes. We further demonstrate the generality of our strategy by applying it to tubulin-binding natural products to modulate their tubulin polymerization activities.

Antimicrobial resistance (AMR) bacteria are spreading worldwide, and there is an urgent need to develop new antibacterial agents that are effective against resistant bacteria[1,2]. In order to develop antibacterial agents that are effective against drug-resistant bacteria such as methicillin-resistant *Staphylococcus aureus* (MRSA), vancomycin-resistant enterococci (VRE), it is important to develop a new class of antibacterial agents that do not interfere with existing resistance mechanisms. Phospho-*N*-acetylmuramoyl-pentapeptide-transferase (MraY) is a bacterial transmembrane enzyme, which is responsible for the formation of lipid I during peptidoglycan biosynthesis (Supplementary Fig. 1)[3,4]. Since MraY is universally present in bacteria and an essential enzyme for their survival[5], it is expected to be a promising

target in antibacterial drug discovery. Many nucleoside antibiotics[6], including tunicamycins[7], muraymycins[8], mureidomycins[9], and capuramycin[10], are known as MraY inhibitors and exhibit antibacterial activity against drug-resistant bacteria (Fig. 1a). These natural products have a uridine moiety as a common substructure, but otherwise are structurally diverse, each exhibiting a characteristic antibacterial spectrum. MraY inhibitory natural products have been extensively studied, and their co-crystal complex structures bound to MraY have been solved and the common uridine moiety binds to the uridine pocket, while other moieties interact with various hot spots (HSs) of MraY (Fig. 1a, b)[11–13]. These studies provide a foundation for structure-based approaches to design improved MraY inhibitors. However, it is

still difficult to rationally design inhibitors because MraY is a conformationally dynamic enzyme. Although no clear interaction pockets with an inhibitor are found in its apo form, MraY undergoes induced fits upon inhibitor binding. Consideration of membrane permeability is also critical because the catalytic site of MraY resides on the cytoplasmic side of the cytoplasmic membrane, thus MraY inhibitors must be able to penetrate the cytoplasmic membrane[11]. Simultaneous

optimization of MraY inhibition and bacterial accumulation is essential to obtain whole-cell activity; however, balancing the requisite physicochemical properties needed for accumulation while retaining molecular features for MraY binding defined by the structure-activity relationship (SAR) is especially challenging given the complex and polar nature of these antibiotics. Furthermore, synthesizing a sufficient number of analogues for structural optimization is not easy due

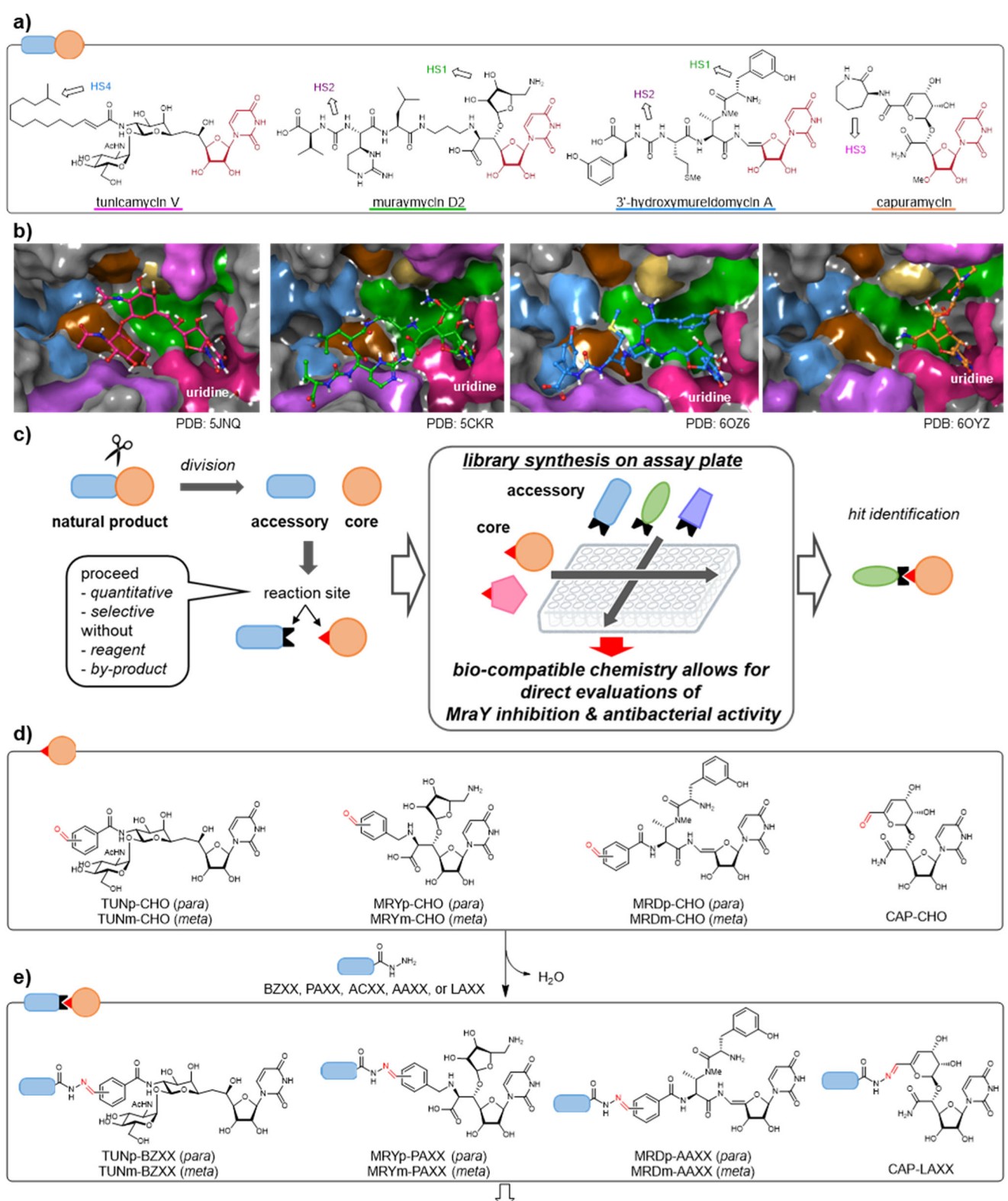

**Fig. 1 | Development of antibacterial agents based on MraY inhibitory natural products with build-up library synthesis strategy. a** Structures of MraY inhibitory natural products have common uridine moiety binding to uridine pocket (red) and other motif with various binding mode in MraY. HS1–4 represent binding hot spot (see details in reference 13, magenta; uridine pocket, green; HS1, purple; HS2, salmon; HS3, cyan; HS4, light brown; HS5, brown; HS6). **b** Complex structures of these antibiotics bound to MraY. Each carbon colour represents each antibiotic in panel a (salmon; tunicamycin, green; muraymycin D2, cyan; 3'-hydroxy-mureidomycin A, orange; capuramycin). **c** Overview of a comprehensive in situ evaluation of the build-up library. (1) Natural products are divided into the core and accessory. (2) Reaction site is introduced into both pairs. The reaction ideally

proceeds quantitatively and selectively without toxic reagents and by-products. (3) The core and accessory fragments are ligated on the assay plates. The resulting library is directly evaluated by enzymatic and cell-based assays. (4) A comprehensive SAR is obtained and hit analogues are identified. **d** Core fragments are the core substructures containing uridine from these antibiotics and formyl group attached to them. Each antibiotic type is represented by three capital letters (tunicamycin; TUN, muraymycin; MRY, 3'-hydroxy-mureidomycin; MRD, capuramycin; CAP) and an additional letter (*para* or *meta* substituted formyl group; p-CHO or m-CHO). Aldehydes are ligated with hydrazine (named BZXX, PAXX, ACXX, AAXX, or LAXX; XX is a number). **e** The name of the obtained hydrazone should be indicated with the aldehyde name in front and the hydrazine name behind (e.g. TUNp-BZXX).

to their complex structure requiring multi-step synthesis. Even with a single natural product, the synthesis of natural product analogues of a large and complicated chemical structure requires multiple steps, accompanied by complicated purification and structure determination processes, leading to tremendously high costs. Therefore, synthesizing a library of natural product analogues as quickly and comprehensively as possible is one of the key challenges in medicinal chemistry based on natural products.

In this work, we develop the platform simplifying comprehensive analogue synthesis of a series of natural products, which accelerates the structural optimization of MraY inhibitory natural products (Fig. 1c). We focus on in situ screening, where a library of compounds is synthesized on an assay plate and biological activity is evaluated without purification[14–17]. Not only can a large number of analogues be rapidly synthesized by this method, but also the amount of compounds required for biological activity evaluation should be small. Therefore, we consider this approach to be one of the solutions that would allow an effective and comprehensive optimization of multiple natural products. Our strategy is to first divide the chemical structure of natural products into two fragments; one is a core fragment, which is expected to play a key role in binding to the target, and the other is an accessory fragment that is expected to further modulate binding affinity to the target, selectivity to off-target(s), and most importantly, the disposition properties. These two fragments are ligated to construct a library of natural products prior to biological evaluation, which we call a build-up library. Many studies of in situ screening often retain by-products derived from reagents such as condensing agents in the reaction mixture, and therefore, are limited to biochemical assays[14]. In this study, the core fragment and a library of the accessory fragments are ligated by a reaction, which proceeds with high chemoselectivity and near quantitative yield without any contaminating reagents or by-products. Our approach avoids lengthy multi-step synthesis, purification, and characterization of each compound and enables direct biological evaluation in an enzymatic and a cell-based assay. In this study, we apply this strategy to optimize biological activities of a series of MraY inhibitory natural products simultaneously in order to develop new antibacterial drug leads tackling drug-resistant infections. Using the 7 cores (four classes) and 98 accessory fragments, the build-up library composed of 686-analogues is prepared. Several analogues exhibiting strong MraY inhibitory and antibacterial activity are identified, and further optimization of these analogues leads to the identification of analogue **2**, which is effective against drug-resistant strains and in mouse infection models. We also apply our build-up library synthesis strategy to a class of tubulin-binding natural products, including epothilone, paclitaxel, and vinblastine, demonstrating the versatility of our strategy.

## Results
### Design of the library
It is important to select an appropriate fragment ligation reaction in terms of its yield, chemoselectivity, and generality. For example, the in situ screening strategy[14] using amide bond formation or the Cu-

catalyzed azide-alkyne cycloaddition (CuAAC) has been reported[18,19]. In these reports, several potent inhibitors were identified by evaluating the in vitro enzyme inhibitory activity of the libraries composed of reaction mixtures. However, these reactions are unsuitable for in situ cell-based assay because of contamination of cytotoxic reagents, by-products, and heavy metals. We chose the hydrazone formation reaction as a fragment ligation to evaluate both MraY inhibitory activity and cell-based susceptibility to pathogens in situ. Since the hydrazone formation reaction is a chemoselective reaction between aldehyde/ketone and hydrazine and produces only $H_2O$ as a by-product[20–25], it can be applied to in situ cell-based assays. We designed a hydrazone library based on MraY inhibitory antibiotics to develop new antibacterial agents as follows (Fig. 1d, e). The ligation reaction between the core aldehydes and the hydrazines was conducted on the microplate, and MraY inhibitory and antibacterial activities of the resulting library were directly evaluated. The aldehyde core fragments share a uridine moiety, which is necessary for binding to MraY (Fig. 1a, b)[26–28]. Additionally, to increase the stability of the hydrazones, conjugated aldehydes were used[29,30]. These core aldehydes were prepared according to the previous syntheses (Scheme S1-4)[31–34].

The contribution of the accessory motifs to MraY inhibition is substantial, as the core structures on their bind and inhibit MraY nearly three orders of magnitude weaker than those of the original natural products[27,28,33]. The accessory motifs also greatly impact bacterial accumulation. In the case of muraymycin, long and lipophilic substituents are considered important for membrane permeability and bacterial accumulation[35]. On the other hand, capuramycin does not require long lipophilic substituents and aromatic substituents are effective[26]. In addition, inhibition of GPT (UDP-*N*-acetylglucosamine: polyprenol phosphate translocase) or non-specific cytotoxicity was affected by the structure of the acyl chain in tunicamycin[36,37] and caprazamycin[38]. For all of these aforementioned reasons, preparing hydrazine fragments of diverse chemotypes is important. We thus prepared various simple acyl hydrazides containing aromatic rings (benzoyl-type: BZ and phenyl acetyl-type: PA) and alkyl groups (acyl-type: AC) from commercially available esters or carboxylic acids as well as *N*-acyl aminoacyl hydrazides containing various lengths of acyl group (acetylamino acid: AA and lipid amino acid: LA) and a range of amino acids with diverse side-chain (Fig. 2a and Supplementary Fig. 2).

### Synthesis and biological evaluation of build-up library
We first confirmed that the hydrazone formation using model compounds proceeded by good conversion only concentrating the mixture of aldehyde and hydrazine and the resulting hydrazones were stable under assay conditions for biological evaluations (see details in Supplementary Fig. 3–7). Next, we assembled a comprehensive build-up library from our set of MraY inhibitory natural product aldehyde cores and hydrazide accessory fragments. Hydrazone formation was performed by simply mixing 10 mM DMSO solution of aldehyde core and hydrazine fragments in approximately 1:1 stoichiometry in 96-well plates in a total volume of 31 µL at room temperature without any additives. After 30 min, the DMSO was removed using centrifugal

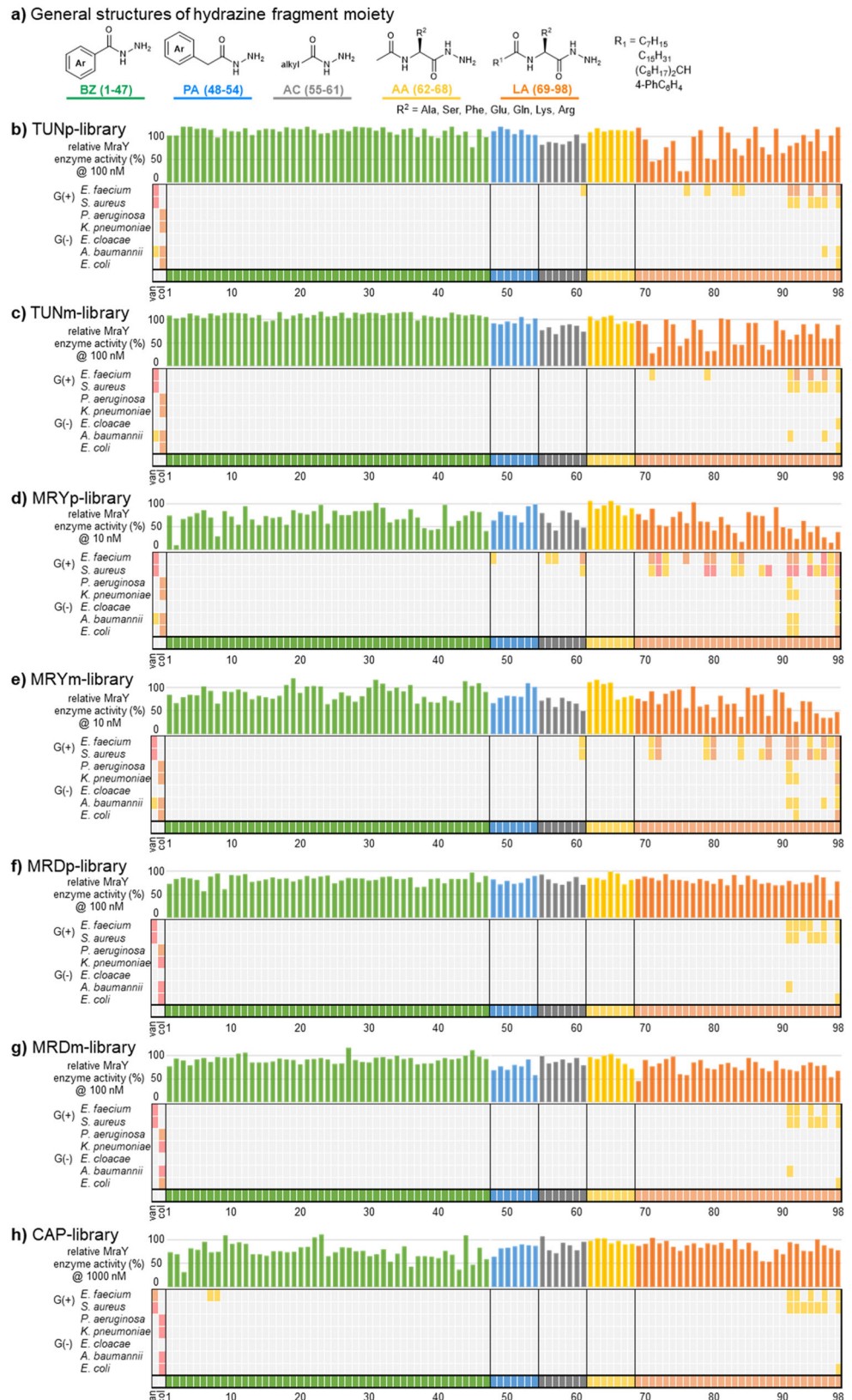

a) General structures of hydrazine fragment moiety

BZ (1-47)  PA (48-54)  AC (55-61)  AA (62-68)  LA (69-98)

$R^1 = C_7H_{15}$, $C_{15}H_{31}$, $(C_8H_{17})_2CH$, 4-PhC$_6$H$_4$

$R^2$ = Ala, Ser, Phe, Glu, Gln, Lys, Arg

b) TUNp-library
c) TUNm-library
d) MRYp-library
e) MRYm-library
f) MRDp-library
g) MRDm-library
h) CAP-library

concentration under vacuum at room temperature overnight, and the resulting residues were dissolved in 30 μL of DMSO to prepare a 5 mM library solution. The LC-MS analysis of the library solution indicated that most hydrazones were obtained at 80% yield or higher, but some reactions depending on the structure of the hydrazine progressed only partially and a few did not proceed at all (Supplementary Fig. 8).

Overall, the hydrazone formation was found to be efficient and suitable for build-up library synthesis because of its clean conversion and simple operation that allowed the preparation of a series of hydrazones. With the build-up library in hand, we first evaluated the MraY inhibitory activity[39-42] of the library (Fig. 2). In subsequent evaluations of the biological activity of the library, the evaluation was performed at

**Fig. 2 | Results of biological evaluations of hydrazone library. a** General structures of hydrazine fragment moiety are shown. Each hydrazine class are represented by different colours (green; BZ, cyan; PA, grey; AC, yellow; AA, orange; LA). **b**–**h** MraY inhibitory activity and antibacterial activity of the hydrazone library. The bar graphs show the relative enzyme activity with DMSO as 100% when the compound is treated at given concentration (b,c; TUN at 100 nM, d,e; MRY at 10 nM, f.g; MRD at 100 nM, h; CAP at 1000 nM. n = 1), with the lower bars indicating higher inhibitory activity. The colour of the bars corresponds to Fig. 2a. Antibacterial activity was evaluated by the minimum inhibitory concentration (MIC) at three points, 0.5, 5, and 50 μM ($n = 1$), and shown as a heat map with the compound on the horizontal axis and the bacterium on the vertical axis. The redder the heat map, the lower the MIC value, i.e., the higher the antibacterial activity (pink; <0.5 μM, orange; 5 μM, yellow orange; 50 μM, light grey; >50 μM). Data of control compounds (van: vancomycin, col: colistin) are shown left column in each panel. *E. faecium* ATCC 35667, *S. aureus* ATCC 29213, *P. aeruginosa* ATCC 27853, *K. pneumoniae* ATCC 13883, *E. cloacae* ATCC 13047, *A. baumannii* ATCC 19606, *E. coli* ATCC 25922.

concentrations assuming 100% conversion. We confirmed that the MraY inhibitory activity of aldehyde cores was 100 ~ 1000 times lower than that of the original natural products (Supplementary Fig. 9)[32–35]. However, in the case of capuramycin aldehyde core, the inhibitory activity did not decrease significantly as reported previously[20,34]. Based on these results, we determined the evaluated concentration of the library so that the MraY inhibition of the aldehyde core would be less than 20%. Also, the MraY inhibition of hydrazine fragments were up to 58%, even at 200 μM. The chemotype of hydrazine moieties contributing to the MraY inhibition differed among the classes of natural products. Namely, the LA-type hydrazine enhanced MraY inhibition in the TUN and MRY sub-library, while the effect of the residue differed among them; in the TUN sub-library (Fig. 2b, c), hydrazine possessing Ala, Phe, and Ser are effective (LA71, LA72, LA75, LA76, LA79, and LA80), but in the MRY sub-library (Fig. 2d, e), basic residue Lys and Arg are more effective (LA92 and LA96)[35,43]. Additionally, some BZ-type hydrazines enhanced MraY inhibition as well as LA-type in the MRY sub-library. The BZ-type hydrazine in CAP sub-library and only LA97 in the MRD sub-library enhanced MraY inhibition (Fig. 2f-h). Overall, the MRY analogues tend to be the most active among the whole libraries. Then, the antibacterial activity of the library was evaluated against the six bacterial pathogens that are important for nosocomial infections with antimicrobial resistance, ESKAPE[44,45]; *Enterococcus faecium, Staphylococcus aureus, Klebsiella pneumoniae, Acinetobacter baumannii, Pseudomonas aeruginosa*, and *Enterobacter cloacae* (Fig. 2). In TUN and MRY sub-library, some hydrazones exhibited moderate or strong antibacterial activity at 0.5 or 5 μM (Fig. 2b–e). These hydrazones have in common that they were of the LA type, with other types showing little antibacterial activity. Furthermore, some analogues in the MRY sub-library were effective against gram-negative bacteria. Hydrazones in MRD and CAP sub-library exhibited no or little antibacterial activity (Fig. 2f–h). In terms of MraY inhibitory activity as well as antibacterial activity, the MRY analogues tend to be the most active among the whole libraries. For more details, see Supplementary Information.

Subsequent studies focused on the MRY sub-library given its particularly potent MraY inhibition and antibacterial activity. Namely, we selected MRYp-BZ2, MRYp-AA63, and MRYp-LA89 as a negative series since these compounds displayed no antibacterial activity, but did possess potent MraY inhibition, and MRYp-LA80, MRYp-LA92, and MRYp-LA98 as a positive series since these compounds displayed both potent MraY inhibition and antibacterial activity (Fig. 3). The compounds were re-synthesized and their chemical structures were confirmed to be identical to the corresponding hydrazones from the MRY sub-library by LC-MS analysis as well as by $^1$H NMR spectra (Supplementary Fig. 11-12). Next, the MraY inhibition of these re-synthesized hydrazones was evaluated (Table 1, Supplementary Fig. 13). Purified hydrazone MRYp-BZ2, MRYp-LA80, MRYp-LA92, and MRYp-LA98 showed potent single-digit nanomolar $IC_{50}$ values, and MRYp-AA63 and MRYp-LA89 exhibited comparatively weaker activity with $IC_{50}$ values ranging from 67, 120 nM, respectively. These results were concordant with semi-quantitative data obtained from the in situ screening of the build-up library. Then, the antibacterial activity of these hydrazones was evaluated (Table 1). Among the hydrazones MRYp-BZ2, MRYp-LA80, MRYp-LA92, and MRYp-LA98 displaying potent MraY inhibition, MRYp-BZ2 exhibited very weak activity against *S. aureus* and *E. faecium* compared to the other three LA-type

hydrazones, which indicates that the long lipophilic chain is important for antibacterial activity. Hydrazone MRYp-LA92 and MRYp-LA98 bearing the basic lysine residue showed a broad spectrum of antibacterial activity and MRYp-LA98 displayed the overall lowest minimum inhibitory concentrations (MICs) ranging from 1–16 μg/mL against the entire ESKAPE pathogen panel.

## Design of stable analogues and evaluating their biological activity

Having verified the comprehensive in situ evaluation of the build-up library, we then proceeded to synthesize stable derivatives because of concern about the inherent susceptibility to hydrolysis of hydrazones during in vivo biological evaluation. Based on the MRYp-LA92 hydrazone, which exhibits a broad spectrum of antibacterial activity, the amide analogues **1**–**4** and the anilide analogues **5**–**8** were designed as stable analogues (Fig. 4). These analogues possess Lys or Arg residue and various linkers connecting the core and the accessory moieties to gain chemical or metabolic stability. These compounds were synthesized in a manner similar to the synthesis of the MRY-type cores (Scheme S5). The analogues **1**–**8** exhibited potent MraY inhibition ($IC_{50}$ 1.7–6.0 nM) and were equipotent to the corresponding hydrazones, indicating that conversion of hydrazone bond to amide bond did not impact MraY binding and inhibition (Table 1, Supplementary Fig. 14). The antibacterial activity of each of these analogues was also similar, paralleling their similar inhibition of MraY. These analogues exhibit particularly strong antibacterial activity against *S. aureus* and *E. faecium* with MIC values of 0.5–1 and 0.25–1 μg/mL, respectively, and excellent activity against several gram-negative pathogens including *K. pneumoniae, A. baumannii*, and *E. coli* with MIC ranging from 4–8, 2–4 and 4–8, respectively. On the other hand, we did observe substantial differences in activity against *P. aeruginosa* where the MIC ranged from 8–128 μg/mL. Finally, when evaluating cytotoxicity against HepG2 cells, all analogues exhibited moderate cytotoxicity ($IC_{50} = 12$–25 μM, Supplementary Table 2). These values did not indicate strong toxicity compared to known MraY inhibitors[35,37,38].

Since these stable analogues are effective against ESKAPE pathogens, especially gram-positive bacteria, the effectiveness of these analogues against antimicrobial drug-resistant strains (methicillin-resistant *S. aureus*; MRSA, vancomycin-resistant enterococci; VRE) and the clinical isolates was investigated (Supplementary Table 1). Analogues **1**–**8** exhibited high efficacy against the reference *S. aureus* and *enterococci* ATCC strains and most of the clinical isolates, even some isolates exhibited resistance to multiple antimicrobial agents, ampicillin, vancomycin, and/or levofloxacin (Supplementary Table 1). These results suggested that these MraY inhibitors are likely not cross-resistance to known drugs evaluated in this study because of their novel mode of action.

## Antibacterial properties of 2 and evaluation of activity in vivo

Next, we evaluated the properties of MraY inhibitors as an antibacterial agent. Analogue **2** was selected to elucidate whether it is bactericidal or bacteriostatic and whether drug-resistant strains emerge (Fig. 5a, b). Analogue **2** reduced colony counts of *S. aureus* by approximately 2 $\log_{10}$ colony formation unit (cfu)/mL after 6 h (4×MIC). In addition, a rapid decrease in colony counts was observed against *E. coli* (under 1 $\log_{10}$ cfu/mL within 3 h treating with 4×MIC). These results indicate

**Fig. 3 | Structures of resynthesis hydrazones.** Hydrazone analogues are summarized along with biological activities.

## Table 1 | MraY inhibitory and antibacterial activity against ESKAPE pathogens

| | IC$_{50}$ for MraY$_{SA}$ (nM) | MIC (µg/mL) | | | | | | |
|---|---|---|---|---|---|---|---|---|
| | | *S. aureus* ATCC 29213 | *E. faecium* ATCC 35667 | *P. aeruginosa* ATCC 27853 | *K. pneumoniae* ATCC 13883 | *E. cloacae* ATCC 13047 | *A. baumannii* ATCC 19606 | *E. coli* ATCC 25922 |
| MRYp-BZ2 | 2.4 | 128 | 64 | >128 | >128 | >128 | >128 | >128 |
| MRYp-AA63 | 67 | >128 | >128 | >128 | >128 | >128 | >128 | >128 |
| MRYp-LA89 | 120 | >128 | >128 | >128 | >128 | >128 | >128 | >128 |
| MRYp-LA80 | 7.4 | 2 | 0.5 | >128 | >128 | >128 | >128 | >128 |
| MRYp-LA92 | 5.0 | 1 | 1 | >128 | 16 | 64 | 8 | 16 |
| MRYp-LA98 | 7.8 | 1 | 2 | 16 | 4 | 8 | 8 | 4 |
| **1** | 6.0 | 0.5 | 0.25 | 8 | 4 | 8 | 4 | 4 |
| **2** | 5.9 | 1 | 0.5 | 16 | 8 | 8 | 4 | 4 |
| **3** | 4.0 | 1 | 1 | 32 | 4 | 8 | 4 | 4 |
| **4** | 5.0 | 1 | 0.5 | 32 | 4 | 16 | 4 | 4 |
| **5** | 3.1 | 0.5 | 0.25 | 32 | 4 | 8 | 4 | 4 |
| **6** | 3.7 | 1 | 0.5 | 32 | 4 | 8 | 2 | 4 |
| **7** | 1.7 | 1 | 0.5 | 128 | 4 | 16 | 4 | 8 |
| **8** | 1.8 | 1 | 0.5 | 128 | 4 | 32 | 4 | 8 |
| colistin | | >128 | >128 | 1 | 4 | >128 | 0.5 | 1 |
| vancomycin | | 1 | 1 | >128 | >128 | >128 | >128 | >128 |

MICs were determined by a microdilution broth method as recommended by the CLSI with cation-adjusted Mueller-Hinton broth. Serial two-fold dilutions of each compound were made in appropriate broth, and the plates were inoculated with 5 × 10⁴ CFU of each strain in a volume of 0.1 mL. Plates were incubated at 37 °C for 20 h and then MICs were scored (*n* = 3).

that **2** exhibits bactericidal activity against both *S. aureus* and *E. coli*. The bactericidal effect of **2** is consistent with the known essentiality of the MraY enzyme, whose depletion using a conditional mutant has been shown to result in rapid cell lysis[5,46]. Moreover, the MIC of **2** increased only 2-fold over 30 days of continuous exposure at 1/2×MIC, which indicates that resistant strains are unlikely to emerge (Fig. 5c).

These properties of **2** are ideal for use as an antibacterial agent. In addition, a morphological change of bacterial cells by treatment with analogue **2** was investigated (Supplementary Fig. 15). As a result, abnormal membrane-like structures were found in the cytoplasm when treated with **2**, but not when treated with DMSO or vancomycin. This change is observed when the cell membrane is impaired[47],

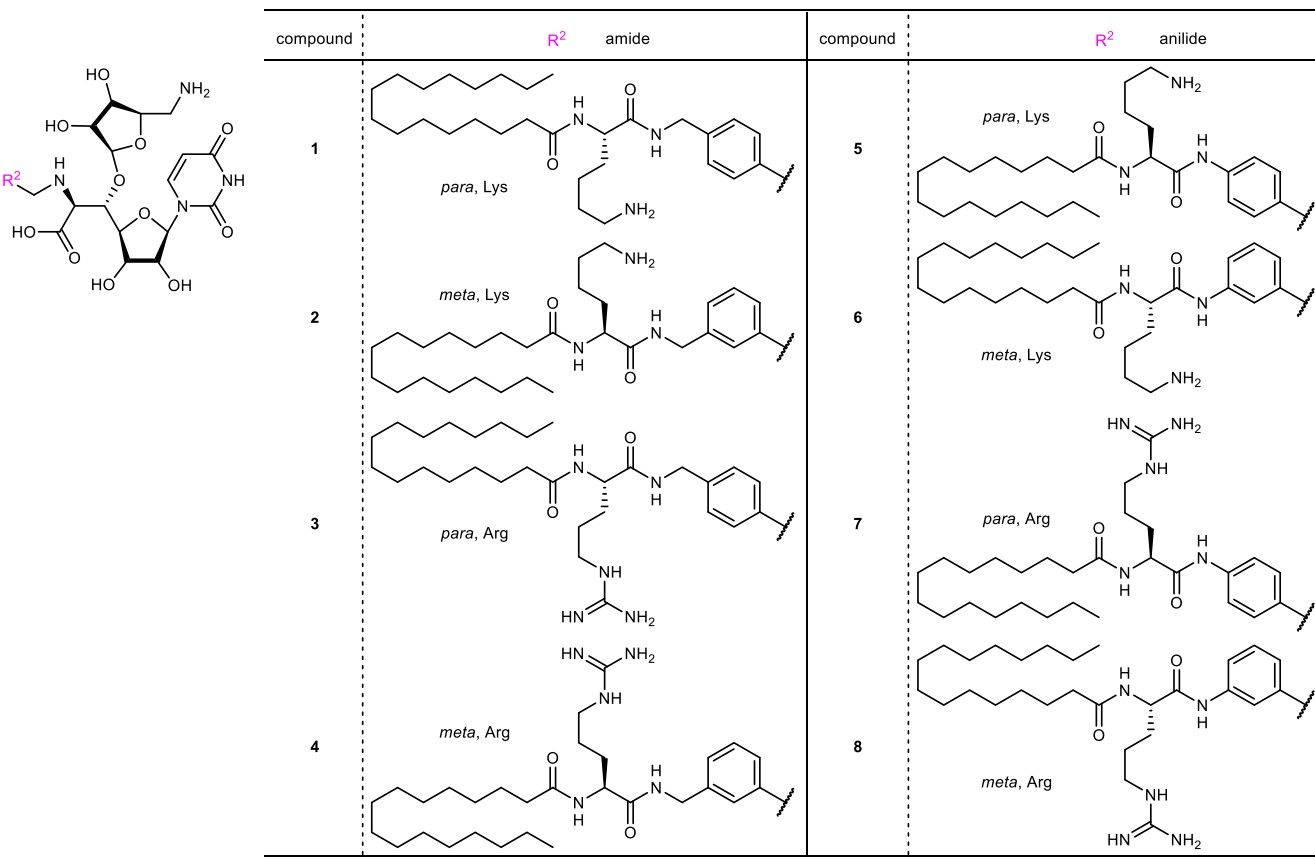

**Fig. 4 | Structures of amide analogues.** Analogue **1**–**4** are amide type analogues connecting core moiety and *N*-acylated amino acid as benzylamide. Analogue **5**–**8** are anilide type analogues connecting core moiety and *N*-acylated amino acid as acyl anilide.

suggesting that analogue **2** has a membrane-disrupting effect in addition to inhibiting peptidoglycan biosynthesis.

Finally, the efficacy of these analogues in a *S. aureus* mouse thigh infection model was investigated (Fig. 5d). An FDA-approved drugs levofloxacin and linezolid were used as positive controls, and both antibiotics when administered at 30 mg/kg reduced bacterial burden by $4\log_{10}$ units in this model compared to the vehicle-treated group. Among the tested analogues, **2** showed the highest efficacy, reducing bacterial burden by $5 \log_{10}$ units when dosed at 30 mg/kg compared to the vehicle-treated group and even showed efficacy at the lower 5 mg/kg dose. Compounds **3** and **6** were also highly effective in reducing bacterial burden by $4\log_{10}$ units, comparable to the positive controls while compound **4** showed little effect even in the 30 mg/kg group. No significant weight loss or other toxicity was observed within the dosing concentrations. Given all of the analogues tested had almost the same antibacterial activity in vitro, we speculate the differences in their in vivo potency in the mouse infection model reflects differences in ADMET properties that influence their pharmacokinetics (PK) and pharmacodynamics (PD) behaviour that subsequently impacts the in vivo exposure at the site of infection. Investigation of the ADMET and PK of these compounds is underway.

## MraY$_{AA}$−2 and MraY$_{AA}$−3 complex structure

We selected analogues **2** and **3**, which exhibits potent inhibition on MraY and promising antibacterial activity, for our structural studies (Table 1). We chose MraY from *Aquifex aeolicus* (MraY$_{AA}$) for our structural studies[11,13,48,49]. MraY$_{AA}$ is an integral membrane enzyme that forms a homodimer. To aid data processing in cryo-EM of this small membrane protein (M.W. ~45 kDa for monomer), we used a previously identified nanobody (NB7), which does not affect MraY$_{AA}$ function, as a fiducial marker[13,48,49]. We determined the cryo-EM structures of the

NB7-MraY$_{AA}$ complexes bound to each of these analogues, which were resolved to 2.88 Å (analogue **2**) and 2.70 Å (analogue **3**) (Supplementary Fig. 16). The ligand densities in both maps are well-resolved, which enable us to model the compounds unambiguously (Supplementary Fig. 17a, b). Our structures revealed that both compounds bind to the site near the cytoplasmic face of the enzyme, which is formed by transmembrane helices (TMs) 5, 8, 9b, and loops C, D, and E (Fig. 6a, b). This binding pocket is highly conserved in terms of sequence (comprised of 34 invariant amino acids) and structure (other nucleoside inhibitors bind to this site) (Supplementary Fig. 17c)[12,13,48,49]. Detailed views of the binding sites for the analogues **2** and **3** reveal extensive contacts between these analogues and MraY$_{AA}$ (Fig. 6c, d). We previously divided the energetically important and highly conserved inhibitor binding site into six hot spots (HSs) of MraY (Fig. 6e)[13,50]. Each class of MraY inhibitors occupies a unique combination of these HSs, resulting in a distinct mode of MraY inhibition. Interestingly, the analogues **2** and **3** exhibit a distinct mode of inhibition compared to the previously determined inhibitors (Fig. 6f). The uridine moiety of **2** and **3** fits into the highly conserved uridine pocket, which is formed by residues K70, G194, L195, D196 and F262, as seen in other nucleoside natural product inhibitors. The 5′-amino-ribosyl moiety interacts with the uridine-adjacent site (HS1) (Fig. 6c, d). Notably, the lysine on **2** and arginine on **3** extend into the TM4/5 groove space (HS6), which was not occupied by their predecessor muraymycin D2, through the shape complementarity and electrostatic interactions between the positively charged amino acids (Lys and Arg) and the negatively charged TM4/5 groove (Supplementary Fig. 17d). The arginine of **3** interacts with the backbone of I128 while lysine of **2** does not have specific interaction with MraY$_{AA}$ (Fig. 6a-d), and these basic amino acids appear to anchor these analogues such that their lipophilic side chains can interact with the hydrophobic groove (HS4), thus increasing the potencies of these

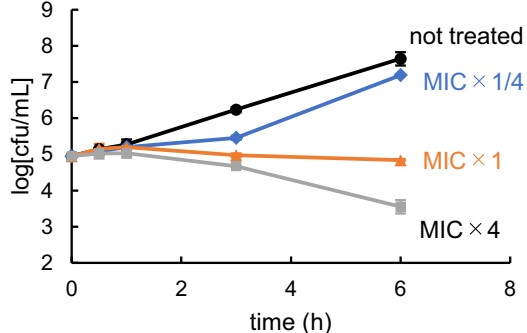

**a)** Time-killing curves against *S. aureus*

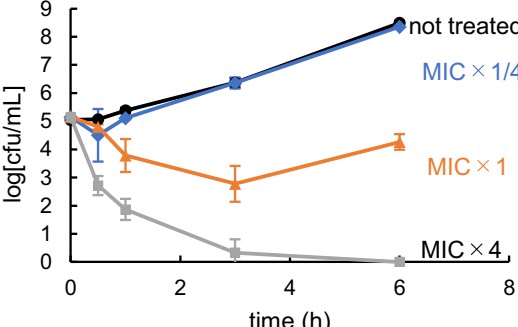

**b)** Time-killing curves against *E. coli*

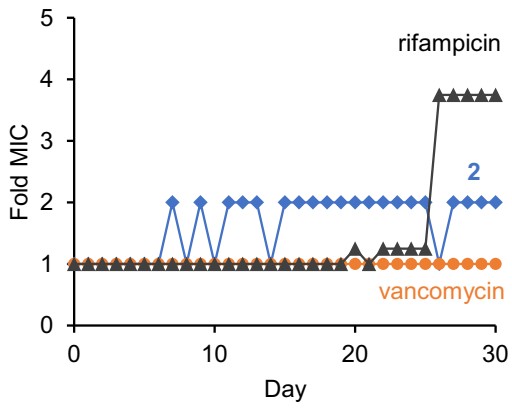

**c)** resistance emergency against *S. aureus*

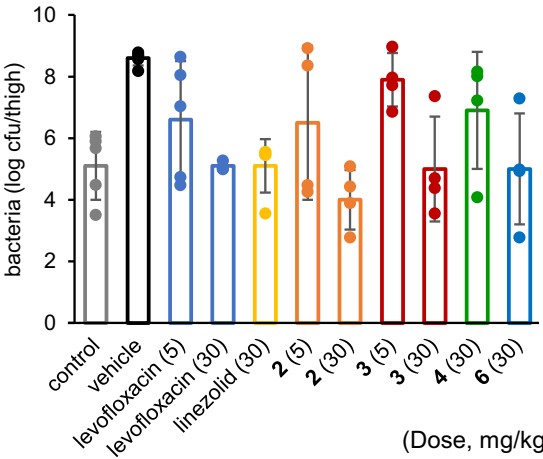

**d)** *S. aureus* mouse thigh infection model

**Fig. 5 | Antibacterial properties of analogue 2.** Time-killing curve against *S. aureus* ATCC 29213 (**a**) and *E. coli* ATCC 25922 (**b**) not treated (black circle) or treated with MIC×1/4 (blue diamond), MIC×1 (orange triangle), and MIC×4 (grey square). These data were collected from three independent experiment (*n* = 3). Error bars represents SD. **c** Resistance emergency plot against *S. aureus* ATCC 29213 treated with MIC×1/2 drugs (rifampicin 0.004 μg/mL (black triangle), vancomycin 0.5 μg/mL (orange circle), or compound **2** 0.5 μg/mL (blue diamond)). **d** The efficacy of analogues against *S. aureus* (ATCC 29213) mouse thigh infection model. The dots represent the raw data, bar graph indicates average values (*n* = 4,5 biologically independent mice). Error bars represents SD. Dose (mg/kg) was shown in parenthesis after the compounds' name or numbers.

analogues. The benzylamide moiety of **2** and **3** extend the molecules into a space different from muraymycin D2, which occupies TM9b/Loop E (HS2) with its urea-dipeptide moiety (Supplementary Fig. 17d). Namely, L191 interacts with the benzylamide moiety of **3** *via* hydrophobic interaction, but otherwise, there are not many specific interactions with MraY, indicating their role in directing the palmitoyl group to towards the lipophilic tunnel lining with residues F180, G184, N187, A188, V296, T299, V302 and I303 (Fig. 6c, d).

Overall, analogues **2** and **3** inhibit MraY in a unique manner compared to previous inhibitors. The core fragments interact with the uridine pocket and HS1 similar to its predecessor muraymycin D2 but these analogues occupy HS6 instead of HS2 (Fig. 6f). Interestingly, the interactions between these analogues and the HS6 of MraY are largely comprised of shape complementarity and electrostatics, without extensive H-bond interactions. However, the interactions with HS6 *via* Arg or Lys play an important role in directing the lipophilic side chains of the analogues to the hydrophobic groove.

## Discussion

A key strength of this approach is the robust chemoselective bioorthogonal chemistry that enables direct evaluation of the in situ synthesized library for biological activity without purification. As a result, we elucidated key SAR trends that influence MraY inhibition and antibacterial activity (Fig. 2). In TUN and MRY sub-libraries

(Fig. 2b–e), the long lipophilic alkyl chain was important for both MraY inhibition and antibacterial activity in alignment with the previous reports, wherein the long lipophilic substituents interact with the lipophilic groove of MraY recognizing undecaprenyl monophosphate ($C_{55}$-P)[11–13,28,43]. This hydrophobic interaction in tunicamycin is critical for the affinity for MraY, and MraY inhibition positively correlated with antibacterial activity (TUNp-LA76, TUNp-LA79). On the other hand, the key interactions with MraY in muraymycin involve the aminoribose and uridine corresponding with the MRYp-CHO and MRYm-CHO. Consequently, the effect of long lipophilic substituents on MraY inhibition was smaller than observed with tunicamycin. However, long lipophilic substituents were necessary for antibacterial activity because the core structure of the muraymycin is highly polar resulting in low membrane permeability and cellular accumulation (MRYp-BZ2 vs. MRYp-LA80/MRYp-LA92). In MRD and CAP sub-libraries (Fig. 2f–h), the long lipophilic substituents were not essential for MraY inhibition. Overall, hydrazones possessing the palmitoyl or isostearoyl group and basic amino acids exhibited the best antibacterial activity and the widest spectrum of activity. We speculated that long alkyl chain acts as an anchor to the bacterial cell membrane and the positive charge of the basic amino acid residue interacted electrostatically with a polar head of the phospholipids[51] in addition to the interaction with MraY, thus explaining the importance of the lipid moiety.

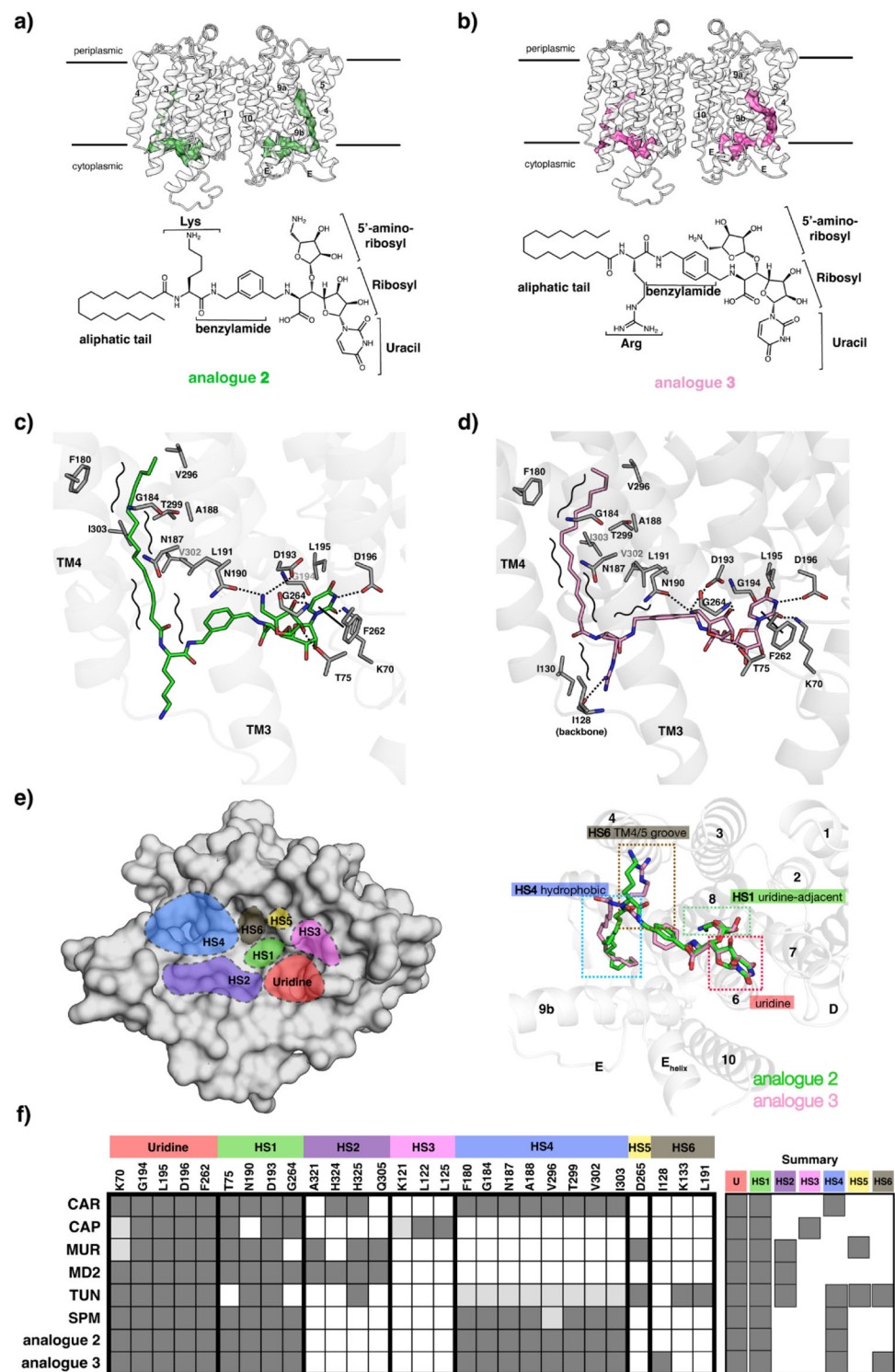

Some hits were identified based on the MRY sub-library, and a hydrazone bond was converted to the stable amide bond. Analogues **1–8** exhibited high MraY inhibition and antibacterial activity, which reflected well on the library results (Table 1 and Figs. 3, 4). These analogues were also effective against drug-resistant strains, and **2** was bactericidal and drug-resistant strains could not be recovered after subculturing for 30 days on a subinhibitory concentration of **2** (Fig. 5, Table 1, Supplementary Table 1). The TEM images suggested that **2** may cause membrane disruption in addition to inhibiting peptidoglycan biosynthesis. This dual action was also observed with polymyxins[52] and aminoglycosides[53,54] and may be one of its promising properties as an

antimicrobial material. Furthermore, **2** was active against *S. aureus* in a mouse thigh infection model, validating our design strategy and further demonstrating the importance of MraY inhibitors as promising antibacterial agents with a novel mode of action.

The selective inhibition of MraY vs. human GPT is a key issue in the development of antibacterial agents. GPT, which belongs to the same superfamily as MraY, is an enzyme that transfers GlcNAc phosphate to dolichol phosphate in the glycan biosynthesis pathway of glycoproteins, and inhibition of human GPT is thought to induce endoplasmic reticulum stress, leading to cell apoptosis[55–57]. Tunicamycin is known to inhibit GPT and is known to be cytotoxic[58–61]. Therefore, in order to

**Fig. 6 | Cryo-EM structures MraY$_{AA}$ in complex with analogue 2 and 3. a** Cryo-EM structure of MraY$_{AA}$ homodimer bound with **2** viewed from the membrane plane (top). The cryo-EM density of **2** is shown in green mesh. Chemical structure of **2** with its substructures labelled (bottom). **b** Cryo-EM structure of MraY$_{AA}$ homodimer bound with **3** viewed from the membrane plane (top). The cryo-EM density of **3** is shown in pink mesh. Chemical structure of 3 with its substructures labelled (bottom). **c** Interactions between **2** and MraY$_{AA}$. Dash line represents hydrogen bonding; Solid line represents π-π stacking; Curved lines represents hydrophobic interaction. **d** Interactions between **3** and MraY$_{AA}$. Dash line represents hydrogen bonding; Solid line represents π-π stacking; Curved lines represents hydrophobic interaction. **e** Structure of MraY$_{AA}$ in surface representation with inhibitor binding site hot spots (HSs) colour-coded and labelled as follows: uridine(red), uridine-adjacent (HS1, lime green), TM9b/LoopE (HS2, purple), caprolactam (HS3, pink), hydrophobic (HS4, cyan), Mg$^{2+}$ (HS5, Gold) and TM4/5 groove (HS6, brown) (left).

The binding sites recognized by **2** and **3** include the uridine (red), uridine-adjacent (lime green), TM4/5 groove (brown) and hydrophobic (blue) pockets. **2** is coloured in green and **3** is coloured in pink (right). The colour scheme is consistent throughout this figure. **f** A barcode representing the interactions each nucleoside inhibitor makes with HS1–6. The residues shown underneath each HS label are found at that site in MraY. Amino acid residue numbering is shown for MraY$_{AA}$. Each row represents a different compound: carbacaprazamycin (CAR), capuramycin (CAP), 3′-hydroxymureidomycin A (MUR), muraymycin D2 (MD2), tunicamycin (TUN), sphaerimicin-1 (SPM), analogue **2**, and analogue **3**. A dark grey square represents an interaction between the corresponding inhibitor and residue. A white square indicates that no contact is made. Squares coloured light grey represent that either the amino acid residue side chain or the inhibitor substructure is not resolved in the structure, but likely makes the relevant binding interaction.

develop MraY inhibitors as antibacterial agents, attention should also be paid to cytotoxicity. Fortunately, analogues **1**-**8** exhibit weak cytotoxicity against HepG2 cell lines and did not exhibit significant toxicity in a mouse model. However, since a wider selectivity index is better for prospective use as an antibacterial agent, it is expected that the search for hydrophobic substituents that can reduce toxicity using this strategy will lead to better antibacterial drug leads.

Intriguingly, thorough our cryo-EM structural studies, we showed that these MraY-analogue complexes (**2** and **3**) share a common interaction pattern with natural MraY inhibitors in uridine and aminoribose moiety, but show different interaction patterns at other sites, suggesting that these analogues are types of MraY inhibitors with different interaction profiles that have not been observed before (Fig. 6). These results underscore the power of our strategy in generating unprecedented types of potent MraY inhibitors. The unique pattern of interactions with MraY stem from the fact that **2** and **3** are comprised of the architecture of the muraymycin core fragment linked to a hydrophobic aliphatic tail and a basic amino acid (either Lys or Arg). This simple compound architecture generates a unique interaction pattern with MraY. In these interaction networks, we suggest that the basic amino acid plays a role in the interactions with HS6, which in turn directs the aliphatic tail groups of the analogues to the hydrophobic groove of MraY. However, it is also possible that these basic residues, in addition to the suggested role in MraY interactions, aid in the permeability of these analogues into the gram-negative bacteria for the reason mentioned in the previous paragraph. The antimicrobial resistance (AMR) pandemic, fuelled in part by the lack of new antibacterial agents to combat drug-resistant pathogens is predicted to be leading cause of mortality by 2050, exceeding deaths caused by all cancers[2]. Consequently, the development of new-generation antibacterial drugs is a pressing matter that requires sustained investment in basic research.

Although there are examples in the literature of the synthesis of a large number of natural product analogues using solid-phase synthesis, our strategy allows for easier preparation of a library of natural product analogues because of only mixing the two fragments at the final synthetic step. Additionally, the library can be easily expanded by simply adding accessory fragments with relatively simple structures, which leads to faster structural optimization based on natural products. This strategy not only simplifies and accelerates SAR research in natural product drug discovery but also reduces resources since even small amounts of cores can be used to obtain SARs. In fact, through the comprehensive in situ evaluation of the build-up library, about 5 μmol of each core was sufficient for LC-MS analysis to confirm purity and product formation, biochemical evaluation against recombinant MraY, and antibacterial activity against seven bacterial species. To the best of our knowledge, this is the first approach to apply in situ evaluation of the build-up library to optimize the structure and biological activity of multiple natural products at once. It enables us to prepare a library of natural products simply, increase the size of the library easily, and

proceed with the optimization process very efficiently. We therefore applied our strategy to tubulin-binding natural products, epothilone B, paclitaxel, and vinblastine. Two core aldehydes each (six in total) were designed and synthesized by introducing formyl groups at completely different positions in these natural products (Supplementary Fig. 18). By reacting all 6 core aldehydes with 98 accessory hydrazines, a hydrazone library consisting of a total of 588 analogues was synthesized. And tubulin stabilizing/destabilizing activity and cell growth inhibitory activity of the core aldehydes and the libraries were evaluated (see in detail Supplementary Information). As a result, the tubulin polymerization assay yielded analogues that exhibited higher activity than that of natural products, and several analogues exhibited cytotoxic activity against HCT-116 in cellular systems. In the case of tubulin-binding natural products, caution is needed in interpreting results when the conversion rate of the ligation reaction is low due to the relatively high biological activity of the core aldehyde. This is because high biological activity originating from the core aldehyde may be observed. It is important to verify the purity of candidate analogues that are judged to have high activity using techniques such as LC-MS. The details of the derivatives obtained from tubulin-binding natural products and additional applications will be discussed in another paper. These results suggest that our strategy can be widely applied to middle size molecules with complex structures. In addition, our strategy has the potential to be applied not only for improving biological activity but also for rapid conjugation with functional molecules.

## Methods

This study complies with all relevant ethical regulations.

### Preparation of core-aldehydes, hydrazines, and stable analogues
See Supplementary Information.

### Preparation of hydrazone library
10 mM DMSO stock solutions of aldehyde and hydrazine were prepared. 16 μL of hydrazine solutions were dispensed into 96-well V-bottom plates [3363, Corning], and 15 μL of aldehyde solutions were added. After shaking the mixtures for 30 min, DMSO was removed under reduced pressure with a plate centrifuge. When the concentration was completed, 30 μL of DMSO was added to each well and the solids were dissolved by shaking to make 5 mM hydrazone solutions. The *N*-octanoyl hydrazones shown in Supplementary Fig. 3-7 were prepared using microtubes instead of well plates.

### LC-MS analysis of hydrazones
5 mM DMSO solutions of hydrazone were diluted with MeCN to prepare 50 μM solutions. These solutions were analyzed according to the following conditions. Instruments: Prominence-i LC-2030C Plus, LCMS-8040 (Shimadzu). Column: J'sphere ODS-M80, 150 × 4.6 mm.D.,

S-4 µm, 8 nm. Sample folder: 15 °C. Injection volume: 10 µL. Flow rate: 0.4 mL/min, Column oven: 30 °C. Detector: UV (254 nm). Solvent: A; 0.1% aqueous formic acid, B; MeCN, 5% to 90% B in A gradient elusion over 15 min. MS method: ESI method; $m/z$ 100-1,500, nebulizer gas; 3 L/min, DL temperature; 250 °C, heat block temperature; 400 °C, drying gas; 15 L/min.

### Resynthesis of hit hydrazones
A solution of MRYp-CHO (1.0 equiv.) and acyl hydrazide (1.05 equiv.) in 0.1% TFA/MeOH (500 µL) was stirred at room temperature for 24 h. After evaporation, the residue was filtered through SepPak-C18 (1 mM TFA in MeCN/H$_2$O; 0%, 20%, 40%, 60%, and 80%. ice-cold), and the fractions containing the hydrazone were collected with freeze-drying. The obtained hydrazones were characterized with $^1$H NMR and LC-MS analysis to avoid hydrolysis under long measurement times such as $^{13}$C NMR measurement.

### Evaluation of MraY inhibitory activity of the hydrazone library
Using the 5 mM DMSO solutions of the library, DMSO solutions for the MraY inhibition assay were prepared (500 nM for MRY, 5 µM for TUN and MRD, and 50 µM for CAP sub-library). Reactions were carried out in a 384-well microplate. 14.6 µL of a solution containing 13.7 µM UDP-MurNAc-dansylpentapeptide and 68.5 µM undecaprenyl phosphate (C$_{55}$-P) (final concentration; 10 and 50 µM, respectively) in an assay buffer [50 mM Tris-HCl (pH 7.6), 50 mM KCl, 25 mM MgCl$_2$, 0.2% Triton X-100, 8% glycerol] was added to each well. 0.4 µL of a hydrazone solution was added to the above reaction mixture. The reaction was initiated by the addition of *S. aureus* MraY enzyme in the assay buffer (5 µL, 11 µg/mL). After 3 h of incubation at 25 °C, the formation of dansylated lipid I was monitored by fluorescence enhancement (excitation at 355 nm, emission at 535 nm) by using an Infinite M200 microplate reader (Tecan). The inhibitory effects of each compound were determined with the addition of DMSO as a control.

### Evaluation of the antibacterial activity of the hydrazone library
MICs were determined by a microdilution broth method as recommended by the CLSI with cation-adjusted Mueller-Hinton broth II (MHB-II, Becton, Dickinson and Company, USA)[62]. Each compound was prepared in DMSO at 100-fold of the final concentration (5 mM, 0.5 mM, and 0.05 mM). The strains were inoculated with $5 \times 10^5$ cfu/mL in 96-well plates (each 100 µL/well), and 1 µL of hydrazone solutions were added. The plates were incubated at 37 °C for 18 h and then MICs were determined by measurement OD$_{600}$ with Infinite 200 PRO (Tecan) or Spark (Tecan).

### Evaluations of MraY inhibitory and antibacterial activity of re-synthesized hydrazones and stable analogues
The IC$_{50}$ values of compounds against MraY$_{SA}$ were determined by the same method as for the library evaluation ($n = 3$). IC$_{50}$ values were calculated using a nonlinear regression curve fit with a variable slope in GraphPad Prism version 4.0a.

MICs were determined by a microdilution broth method as recommended by the CLSI with MHB-II. For MICs of stable analogues, a 2-fold dilution of each compound was prepared in DMSO at 100-fold of the final concentration. The strains were inoculated with $5 \times 10^5$ cfu/mL in 96-well plates (each 100 µL/well), and 1 µL of DMSO solutions of the samples were added (contained less than 1% DMSO). The plates were incubated at 37 °C for 18 h and then MICs were determined. MICs of stable analogues in Table 1 were the highest value in three trials. MICs in Supplementary Table 1 were determined in one trial.

### Evaluation of toxicity against HepG2 cells
HepG2 cells (ATCC HB-8065) were cultured at 37 °C under 5% CO$_2$ in air in E-MEM (with L-glutamine, phenol red, sodium pyruvate, non-essential amino acids and 1,500 mg/L sodium bicarbonate, wako) supplemented with 10% foetal bovine serum (FBS). Cells were regularly passed to maintain exponential growth. HepG2 cells were sowed by $1.0 \times 10^4$ cells for each well in 96-well plate and cultured at 37 °C under 5% CO$_2$ in air for 24 h. The cells were treated with a solution of test compounds in DMSO at various concentration and incubated at 37 °C under 5% CO$_2$ in air for 24 h. After that, the WST-8 (Dojindo) cell proliferation assay was used. Cell viability was indicated as a percentage of the negative control (cells treated with DMSO only) and that value was fixed at 100%.

### In vitro time-kill studies
The bactericidal activity of **2** against *S. aureus* ATCC 29213 and *E. coli* ATCC 25922 was determined using the time-kill method according to the CLSI guideline. *S. aureus* was grown overnight in a brain heart infusion medium (BHI), then the cells were diluted with MHB-II (approximately $1 \times 10^5$ cfu/mL). The test compound was added to the bacterial cultures at concentrations 0.25, 1, and 4 times the MIC containing less than 1% DMSO, and incubated at 37 °C. At 0, 0.5, 1, 3, 6, and 24 h after the addition of the test compound, samples were collected from the culture medium, serially diluted, spread on cation-adjusted MH-agar, and cultured at 37 °C. The number of colonies growing on the agar plate was counted after 24 h. This experiment was performed in triplicated trials.

### Resistance studies
*S. aureus* ATCC 29213 was inoculated into 1 mL of MHB-II containing sub-minimum inhibitory concentration (sub-MIC) of **2** (0.5 mg/L), vancomycin (0.5 mg/L), or rifampicin (0.004 mg/L), and cultured at overnight at 37 °C. Each 1 uL (approximately, $10^6$ cfu) of the culture was inoculated into fresh MHB-II containing sub-MIC of each regent and cultured overnight at 37 °C. This step was continued for 30 days. MICs of these derived mutants were determined by broth microdilution method according to CLSI guidelines.

### TEM image
*S. aureus* ATCC 29213 strain was used to obtain TEM images. *S. aureus* was grown overnight in MHB (approximately $1 \times 10^9$ cfu/mL). The strain was inoculated in microtubes (750 µL × 3), and then 7.5 µL of the DMSO solutions (DMSO only, 6400 µg/mL of **2**, and 3200 µg/mL of vancomycin) were added (contained less than 1% DMSO). The microtubes were incubated at 37 °C for 3 h. 750 µL of fixing solution A (4% paraformaldehyde and 4% glutaraldehyde in 0.1 M phosphate buffer pH 7.4) was added to the above bacterial culture. The resulting mixture was mixed, and the microtube was allowed to stand sideways at 4 °C for 1 h. After centrifugation, the supernatant was removed, and fixing solution B (2% glutaraldehyde in 0.1 M phosphate buffer pH 7.4) was added to the above pellet. After the pellet was broken up by vortexing, it was fixed at 4 °C. The fixed samples were used for analysis.

### Mouse infection models
Each test compound was prepared in DMSO as a solvent to 18 mg/mL and then diluted with saline. The day before inoculation, the test strain (*S. aureus* ATCC 29213) was inoculated into 20 mL of Muller Hinton Broth and incubated at 37 °C, 200 rpm for 16 h with shaking. The culture medium was centrifuged at $2000 \times g$ for 10 min and the supernatant was removed. The precipitate was resuspended in 20 mL of sterile saline and diluted 300-fold in Muller Hinton Broth to a bacterial concentration of $1 \times 10^6$ cfu/100 µL as the test solution.

For this experiment[63], 55 mice (Strain: Slc:ICR, Sex: Female, Age: 6-week old, Dark/light cycle: 12-h light and 12-h dark cycles, Ambient temperature: $22 \pm 1$ °C, Humidity: $50 \pm 10$%) were used. Four days and one day before inoculation, 150 mg/kg and 100 mg/kg of cyclophosphamide were administered intraperitoneally to mice to induce a state of granulocytopenia. The test solution was inoculated into the thighs of mice under ether anaesthesia at $1 \times 10^6$ cfu/100 µL/thigh. Two

hours after inoculation, the thigh muscle of mice in the control group was removed and the number of viable bacteria in the thigh muscle was determined. The viable bacterial counts were determined by diluting a homogenate of thigh muscle prepared using an ultra-disperser [LK-21, Yamato Scientific] with saline, applying it on an agar medium, and counting the colonies after 24 h of incubation at 37 °C. For the other groups, the test compound (5 or 30 mg/10 mL/kg) was administered subcutaneously on the back 2 h after inoculation with bacteria, and 24 h later, the thigh muscle was removed and the number of viable bacteria in the thigh muscle was determined.

Ethical approval was obtained through the Wakunaga Pharmaceutical Company Institutional Animal Care and Use Committee (Protocol number: 0303). Only female was used in the study according to a previous report.[63]

### Expression, purification, and grid sample preparation of MraY$_{AA}$ in complex with 2 and 3

NB7 and MraY$_{AA}$ were expressed and purified as follows[11,13,50]. NB7 was previously identified as a potent MraY$_{AA}$ binder that recognizes the periplasmic face of the enzyme, away from the cytoplasmic active site[11]. NB7 (Nanobody 7) expression plasmids were transformed into C41(DE3) *E. coli* cells, which were used to inoculate Terrific Broth (TB; Fischer Scientific). The cultures were incubated with shaking at 37 °C until an OD$_{600}$ of 0.5 was reached. Protein expression was induced with 1 mM IPTG and further incubated at 25 °C overnight (~18 h). Cells were then harvested by centrifugation (5500 × $g$, 15 min) and resuspended in buffer containing 50 mM Tris-HCl pH 8, 150 mM NaCl, and 20% sucrose. The resuspended cells were rotated for 30 min at room temperature after which they were centrifuged at 13,000 × $g$ for 10 min. The pellet was retained, rapidly resuspended with ice cold buffer (50 mM Tris-HCl pH 8 and 150 mM NaCl), and rotated for 30 min at 4 °C. The sample was then centrifuged (13,000 × $g$, 10 min, 4 °C) and to the clarified supernatant, 1 mM phenylmethylsulfonyl fluoride (PMSF) and DNase I (20 mg) were added. The mixture was then incubated with cobalt resin (Talon) at 4 °C with rotation for 1 h. The cobalt resin was then collected and washed with 10 column volumes of wash buffer containing 50 mM Tris-HCl, pH 8, 150 mM NaCl and 10 mM imidazole. The protein was eluted with 200 mM imidazole. Nanobodies were further purified by size exclusion chromatography on a Superdex 200 10/300 GL column equilibrated with 50 mM Tris-HCl pH 8 and 150 mM NaCl.

The gene corresponding to MraY$_{AA}$ was synthesized as a fusion with a decahistidine-maltose binding protein (His10x-MBP), which was codon optimized for expression in *E. coli*. A PreScission protease site between MraY$_{AA}$ and His-MBP was introduced. MraY$_{AA}$ was cultured at 37 °C and was induced with 0.4 mM IPTG when OD$_{600}$ reaches 0.8 for 4 h in C41 (DE3) cells. All purification steps were performed at 4 °C. The bacterial cell pellet was resuspended in 50 mM Tris, 150 mM NaCl, 1 µg/mL leupeptin, 1 µg/mL pepstatin, 1 µg/mL aprotinin, DNase I, 1 mM PMSF, and 2 mM β-mercaptoethanol (βME). The His10x-MBP-MraY$_{AA}$ fusion protein was extracted with 30 mM dodecyl-maltoside (DDM, Anatrace) for 2 h, followed by centrifugation at 13,000 × $g$ for 40 min to remove insoluble material. The supernatant was subsequently incubated with cobalt affinity resin (Talon) in the presence of 7 mM imidazole for 1 h. The resin was washed with 20 column volumes of wash buffer (50 mM Tris, 150 mM NaCl, 10 mM imidazole, 1 mM DDM, 2 mM βME) and eluted in the presence of 200 mM imidazole. MraY$_{AA}$ was isolated by cleaving His10x-MBP tag using PreScission protease overnight. MraY$_{AA}$ was purified by size-exclusion chromatography (SEC) with a Superdex 200 10/300 GL column in 20 mM Tris-HCl, 150 mM NaCl, 5 mM decyl-maltoside (DM, Anatrace), and 2 mM DTT. The purified MraY$_{AA}$ was combined with nanobody NB7 at a 1:1.5 molar ratio, and the complex was purified by SEC again to remove excess NB7. The peak fractions containing the MraY$_{AA}$-nanobody complex were harvested, concentrated to 7.7 mg/mL. MraY$_{AA}$-nanobody

complex were combined with 2 or 3 at 1:10 molar ratio of protein to inhibitor on ice for at least 30 min before freezing. 2% of DMSO were added to the samples right before freezing. The samples were plunge frozen using Leica EM GP2 Automatic Plunger Freezer at 4 °C and 90% humidity. A sample volume of 3 mL sample was applied to a freshly glow-discharged UltrAuFoil R1.2/1.3 300 mesh (Quantifoil); the sample was incubated for 1 min and blotted with Whatman No. 1 filter paper for 3 s followed by plunge-freezing in liquid-ethane cooled by liquid nitrogen.

### Data collection

MraY$_{AA}$-NB7-**2** and MraY$_{AA}$-NB7-**3** datasets were collected using a Titan Krios (Thermo Fisher) transmission electron microscope operating at 300 kV equipped with a K3 detector (Gatan) in counting mode with a BioQuantum GIF energy filter (slit width of 20 eV), using the Latitude-S (Gatan) single-particle data acquisition program. Data were collected at a magnification of ×81,000 with a pixel size of 1.08 Å at the specimen level.

For MraY$_{AA}$-NB7-**2** dataset, 2299 movies were collected, and each movie contained 60 frames over 4.6 s exposure time, using a dose rate of 15 e-/pix/s for a total accumulated dose of ~60 e-/Å$^2$. The nominal defocus range was set from −0.8 to −2 μm. For MraY$_{AA}$-NB7-**3** dataset, 2087 movies were collected and each movie contained 40 frames over 2.3 s exposure time, using a dose rate of 30 e/pix/s for a total accumulated dose of ~60 e-/Å$^2$. The nominal defocus range was set from −0.8 to −1.8 μm.

### Data processing

MraY$_{AA}$-NB7-**2**: Beam-induced motion correction and dose-weighing was performed using MotionCor2[64]. Corrected micrographs were then imported into cryoSPARC for contrast transfer function (CTF) estimation with patch CTF[65]. Particle picking was performed with blob picker in cryoSPARC. Particles were then extracted with a 64-pixel 4x binned box size at 4.32 Å per pixel and subjected to two-dimensional (2D) classification. A total of 277,701 particles were selected and re-extracted unbin at 1.08 Å per pixel which then subjected to cryoSPARC 1-class ab-initio reconstruction. The reconstruction already showed distinct TMs features, which is then subjected to one round of non-uniform refinement[66]. The particles were then transferred to RELION for two arounds of CTF refinement and Bayesian polishing[67], followed by non-uniform refinement and local refinement with C2 symmetry applied. The particles were then transferred back to RELION for the third round of CTF refinement and Bayesian polishing, followed by non-uniform refinement and local refinement, which yielded a reconstruction to 2.88-Å resolution (Supplementary Fig. 16e). Map were sharpened with a B-factor of −30.

MraY$_{AA}$-NB7-**3**: Beam-induced motion correction and dose-weighing was performed using MotionCor2[64]. Corrected micrographs were then imported into cryoSPARC for contrast transfer function (CTF) estimation with patch CTF[65]. Manually curate to select 1797 micrographs. Particle picking was performed with blob picker in cryoSPARC. Particles were then extracted with a 64-pixel 4x binned box size at 4.32 Å per pixel and subjected to two-dimensional (2D) classification. A total of 274,072 particles were selected and re-extracted unbin at 1.08 Å per pixel which then subjected to cryoSPARC 1-class ab initio reconstruction. The reconstruction already showed distinct TMs features, which is then subjected to one round of non-uniform refinement[66]. The particles were then transferred to RELION for CTF refinement and Bayesian polishing[67]. Then C2 symmetry applied followed by another round of CTF refinement, Bayesian polishing, followed by non-uniform refinement and local refinement. To improve ligand density, the particles were transferred back into RELION for 3D classification (k = 3, t = 32). 68364 particles set were selected for another two rounds of CTF refinement, Bayesian polishing followed by non-uniform refinement and local refinement, which

yielded a reconstruction to 2.70-Å resolution (Supplementary Fig. 16j). Map were sharpened with a B-factor of −30.

## Model building, refinement and alignment

The model-building process is similar for both structures reported. A previously published MraY$_{AA}$-muraymycin D2 structure (PDB 5CKR) was used as a reference. The placement of MraY$_{AA}$ and nanobody was performed by rigid body fitting and the structures were manually refined using real-space refinement in Coot with ideal geometry restraints[68]. The restraints for **2** and **3** were calculated in Elbow (as implemented in Phenix[69]) from isomeric SMILES strings. Bond lengths and angels were then inspected and adjusted manually to ensure correct stereochemistry before being fitted into the cryo-EM maps in Coot. The MolProbity[70] server (http://molprobity.biochem. duke.edu) was utilized to identify problematic regions in the models, which were then manually adjusted in Coot. The final refinement was performed using the phenix-real_space_refine function with global minimization and secondary structure restraints as implemented in the Phenix suite[69]. Structural analyses and illustrations were performed using PYMOL (Schrödinger) and UCSF Chimera X[71]. Data collection and refinement statistics are provided in Supplementary Table 4.

## Reporting summary

Further information on research design is available in the Nature Portfolio Reporting Summary linked to this article.

## Data availability

Data supporting the findings of this manuscript are available from the corresponding author upon request. Details about characterization data of synthetic organic compounds and experimental procedures are available in Supplementary Information. The coordinates generated in this study have been deposited in the Protein Data Base under accession code 9B70 (MraY-Analogue **2**) and 9B71 (MraY-Analogue **3**). The cryo-EM density maps generated in this study have been deposited in EMDB under accession code EMD-44293 (MraY-Analogue **2**) and EMD-44294 (MraY-Analogue **3**). Source data for Fig. 2, Fig. 5 and Table 1 and Supplementary Fig. 9, Figure 13, Figure 14, Figure 18-22, Table 1, and Table 2 can be found in Source Data. Source data are provided with this paper.

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

## Acknowledgements

We are thankful to Dr. K. Nishiguchi and Dr. S. Arioka (Shionogi Co., Ltd.) for MraY expression and purification, and Tokai Denshikenbikyo Kaiseki for TEM image photography. We thank Dr. Suo Yang for cryo-EM data collection and help with cryo-EM data processing and Dr. Nilakshee Bhattacharya at SMIF for assistance with the microscope operation. Cryo-EM data was collected at Duke University Shared Materials Instrumentation Facility (SMIF). This research was supported in part by JSPS KAKENHI Grant-in-Aid for Scientific Research (B) (Grant Number

22H02738 and 19H03345 to S.I., 21H03622 to T.S.), Grant-in Aid for Scientific Research on Innovative Areas "Frontier Research on Chemical Communications" (No 18H04599 and 20H04757 to S.I.), Grant-in-Aid for Research for Young Scientist (Grant Number JP19K16648 to T.S.), Grant-in-Aid for Research Activity Start-up (Grant Number 22K20704 to K.Y.), Takeda Foundation, The Tokyo Biomedical Research Foundation and was partly supported by Hokkaido University, Global Facility Centre (GFC), Pharma Science Open Unit (PSOU), funded by MEXT under "Support Program for Implementation of New Equipment Sharing System", Platform Project for Supporting Drug Discovery and Life Science Research (Basis for Supporting Innovative Drug Discovery and Life Science Research (BINDS)) from AMED under Grant Number JP18am0101093j0002 and JP22ama121039, AMED-CREST, AMED under Grant Number JP23gm1610012 and JP23gm1610013 to S.I., AMED under Grant Number JP19ak0101118h0001, AMED under Grant Number 21ak0101118h9903 to T.S., JST START Program: ST211004JO to T.S., Japan Initiative for Global Research Network on Infectious Diseases (J-GRID) from the Ministry of Education, Culture, Sport, Science, and Technology in Japan, MEXT for the Joint Research Program of the Research Centre for Zoonosis Control, Hokkaido University, and the Duke Science Technology Scholar Fund (S.-Y.L.).

## Author contributions

K.Y. and S.I. designed the research and K.Y., T.S., S.T., M.H., S.Y., and S.I. designed the experiments. K.Y., K.A., R.K., S.K., and R.R.R. synthesized compounds. K.Y. and K.A. performed the MraY assay. T.S. performed in vitro antibacterial assay. D.K. and K.S. performed an antibacterial assay in the mouse infection model. A.H. and S.-Y.L. performed the determination of a complex structure with cryo-EM. K.Y. and K.A. performed the synthesis and the evaluation of libraries of tubulin-binding natural products. K.Y., T.S., A.H., K.A., R.K., S.-Y.L., and S.I. wrote the paper. All authors discussed the results, commented on the paper, and approved the final version of the manuscript.

## Competing interests

D.K. and K.S. are Wakunaga employees. The remaining authors declare no competing interest.

## Additional information

[1]Faculty of Pharmaceutical Sciences, Hokkaido University, Kita-12, Nishi-6, Kita-ku, Sapporo 060-0812, Japan. [2]Center for Research and Education on Drug Discovery, Faculty of Pharmaceutical Sciences, Hokkaido University, Kita-12, Nishi-6, Kita-ku, Sapporo 060-0812, Japan. [3]Laboratory of Veterinary Hygiene, School/Faculty of Veterinary Medicine, Hokkaido University, Kita-18, Nishi-9, Kita-ku, Sapporo 060-0818, Japan. [4]Graduate School of Infectious Diseases, Hokkaido University, Kita-18, Nishi-9, Kita-ku, Sapporo 060-0818, Japan. [5]One Health Research Center, Hokkaido University, Kita-18, Nishi-9, Kita-ku, Sapporo 060-0818, Japan. [6]Department of Biochemistry, Duke University School of Medicine, Durham, NC 27710, USA. [7]Drug Discovery Laboratory, Wakunaga Pharmaceutical Co., Ltd., 1624, Shimokotachi, Koda-cho, Akitakata-shi, Hiroshima 739-1195, Japan. [8]Division of Laboratory Medicine, Sapporo Medical University Hospital, Minami-1, Nishi-16, Chuo-ku, Sapporo 060-8543, Japan. [9]Department of Infection Control and Laboratory Medicine, Sapporo Medical University School of Medicine, Minami-1, Nishi-16, Chuo-ku, Sapporo 060-8543, Japan. [10]Department of Microbiology, Sapporo Medical University School of Medicine, Minami-1, Nishi-17, Chuo-ku, Sapporo 060-8556, Japan. [11]Global Institution for Collaborative Research and Education (GI-CoRE), Hokkaido University, Kita-12, Nishi-6, Kita-ku, Sapporo 060-0812, Sapporo, Japan. ✉e-mail: k.yamamoto@pharm.hokudai.ac.jp; ichikawa@pharm.hokudai.ac.jp

