## [Peer Review File · Nature Communications]

Development of a Natural Product Optimization Strategy for inhibitors against MraY, a promising antibacterial targetREVIEWER COMMENTS

Reviewer #1 (Remarks to the Author):

The manuscript Entitled "Development of a Natural Product Optimization Strategy for inhibitors against the cell wall synthesis enzyme *MraY*, a promising antibacterial target" presents a report of a higher-throughput, bioassay-compatible synthetic approach to create analogs for natural products. It puts forward the proposal that such a system would allow more facile analogs to challenging natural products to be created directly on multi well plates which would enable rapid bioassay. The idea is an interesting one.

Overall, the manuscript is well written and the figures and data are well articulated. The authors had success in using a series of structurally closely-related antibiotics as templates for their "build-up" stage of library creation. They then used a simple chemical ligation technique to attach a variety of "accessory motifs" to create their final library for testing (and also to optimize around specific structures). The system worked well and they were able to generate an analog that showed good *in vitro* activity and also was active in an animal model of infection.

Though the technique worked in this instance, this reviewer questions the broad applicability of this technique to other natural product molecules. The authors were advantaged in this study by the the fact that:

1. all their core compounds were closely related structures.
2. X-ray structures of all the bound lead molecules were available prior to initiating synthesis.
3. All their compounds bound to the same target at the same binding site.
4. The common structural core of the starting molecules bound to a well-defined pocket.
5. The bioassay for improved function only relied on cellular permeability in a microbe.
6. The core substructures chosen for the analog synthesis were themselves complex molecules but were readily available.

These are all advantages that most natural products would not have going in to such a study. In addition, due to the ligation strategy the authors used, they then had to synthesize more stable derivatives using a different chemical method. This raises a question of whether or not the plate-based method they used would provide sufficient sample material for more than one assay. Also, the authors quote ~80% yields optimally using their method. 20% contamination with building blocks could result effects that alter assay results (i.e. core structures could still bind to target proteins and reduce efficacy of the fully articulated analog). Finally, the structure that provided the best efficacy was very close structurally to the initial molecule tunicamycin (having the sugar units exchanged). Though a fine result, not particularly non-intuitive as far as analog design by other means.

Overall, this is a very nice paper with good results but this reviewer does not think its board applicability rises to the level of publication in *Nature Communications*. Should the authors be able to show broader utility, and address some of the concerns listed above, I would recommend publication

Reviewer #2 (Remarks to the Author):

The manuscript describes the synthesis of ~700 analogues of natural products that are known to inhibit the enzyme *MraY*. The analogues are tested for their inhibitory activity, antibacterial properties, and for some, cytotoxicity. Two selected hits were then co-crystallized with *MraY*, providing a structural basis for their activity. The content is significant with regard to *MraY* inhibitor design, and the approach has elements of novelty. The conclusions are generally supported by the data. However, as noted below, the syntax, grammar, and general flow make the manuscript difficult to follow and understand. Additional concerns are as follows:

1. Incorrect grammar usage makes the manuscript difficult to follow. The structure of sentences

and paragraphs further detracts from the significance of the study. For example with the latter, a significant portion of the introduction is dedicated to why the authors employed a so-called build-up strategy (which, by the way, is essentially a convergent synthesis strategy) to make natural product analogues. This information could be dramatically shortened to perhaps even one sentence. One potential way to restructure the introduction is first to introduce *MraY*, then describe the known natural product inhibitors (including grouping based on inhibition patterns, potency, and other pertinent information), and finally how a convergent synthesis approach was developed to make simplified analogues that consist of a core component (binding to the "uridine" binding site) and a peripheral moiety (binding to one of the additional hot spots) to make a focused library.

2. On page 9, the authors describe dissolving the crude reaction mixture into DMSO to make a reported 5 mM stock that assumes quantitative conversion of the reagents. However, it is noted that while many were obtained at 80% or higher, some did not have great yields or proceed at all. Was the conversion to the product considered when reporting concentration used in Figure 2 (for example TUN-type library at 100 nM) and Figure 3 for MIC values? How can one product be compared to another if, for example, one reaction gives quantitative conversion while the other 10%? Also with regard to the activity, it was not clear whether the separate fragments (core or peripheral moiety) had activity alone or together without covalent coupling. Finally, it would be nice to have information regarding the IC_{50} , MIC, and spectrum of the natural products as comparators. If they were not used in this study, then citing the appropriate reference and providing the values would suffice.

3. On page 12, the cytotoxicity against HepG2 cells is stated at 38-66 μ M. Is this IC_{50} ? How does the therapeutic index (antibiotic vs toxicity activity) compare when considering the parent natural product? For example, it is well known that tunicamycin is toxic and therefore would be expected to have both antibacterial and cytotoxic activities. Does the ratio increase or decrease with the analogues? The noted cytotoxicity makes the mouse model study concerning. Did the dose that was used have any effect on the mice? Is the maximal tolerated dose known?

4. The statement that the analogues have "different" or "novel" pharmacological profiles likely needs clarification. What is meant by pharmacological profile, and what are the analogues being compared to? If, for example, the authors are attempting to say that the analogues, specifically analogue 2, causes the formation of abnormal membrane-like structures, has the parent natural product (for example muraymycin D1) been examined in the same assay for comparison? Perhaps instead of broadly stating differences in pharmacological profiles, specifically note what property is unique with respect to manifestation of its antibacterial activity. It is worth pointing out that other natural products (polymixins, aminoglycoside, etc) are known to disrupt membrane structure and, for aminoglycosides, bind the ribosome thus having a dual mechanism of action in the manner of what is being proposed for the analogues in this manuscript.

Reviewer #3 (Remarks to the Author):

Review of: Development of a Natural Product Optimization Strategy for inhibitors against the cell wall synthesis enzyme *MraY*, a promising antibacterial target

In this work, Yamamoto et al. create a workflow to generate a bio-compatible library of potential *MraY* antibiotics. They screen this library with both enzymatic and cellular assays to find potential antibiotics. Subsequently they further characterise some of these hits, including two cryoEM structures. This is a nice body of work for a manuscript, though there are some significant issues with the communication of this data, which preclude my suggestion for publication at this time. I believe a significantly revised manuscript, containing the same content, could be suitable for publication in this venue.

Major Comments:

- Figure one has most of the components to be an excellent figure, yet its presentation is a bit confusing. I have a number of suggestions to address this:
 - o Figure 1 realistically has many panels, yet is split into A and B. This can lead to confusion for the reader and in regard to what is what when comparing the figure to the description below. As well, the numbering from A is in the figure legend but not in the figure itself, that would be a great

addition. As well, I would recommend the whole figure be converted to at least A-E for clarity.

o Regarding the order of the content, the figure goes from a schematic overview, then back to the data that supported the rationale for the workflow above, then back to zooming into a section of the workflow. I think it would be most logical to start the figure with the first two panels of B (tunicamycin V, muraymycin D2...) to show the rationale, then proceed to the current A overview, followed by the lower two panels of D.

o The structural overlay could use some improvement, particularly for clarity at its current size.

o Somehow it should be added to this figure that this workflow w/ the bio-compatible chemistry allows for the antibacterial assays (diagrammatically, as the font "direct evaluation of enzymatic and cell-based assay" very much under sells this point). This is a key advance made in this work and should thus be presented as such.

• Some comment on *MraY* similarities to human GPT (GlcNAc phosphotransferase) is required in the introduction. Given tunicamycin can inhibit mammalian N-linked glycosylation, this is important to discuss. Does the workflow and methodology of this work provide any insight into potential toxicities? Have any steps been taken to design a library that will produce hits that avoid toxicity in humans?

o Toxicity not seen for selected compound in HeLa2 cytotoxicity assays and mouse studies – was this by chance or design?

• Figure 2 legend is lacking a majority of detail to fully comprehend the figure. Please add description of all acronyms used, description of the various panels (A-E).

• In Figure 2, vastly different concentrations are used for each "class" of compounds. Were these all tested at 10, 100, 1000 nM? If so, this data should, in the very least, be included in the Extended Data.

• Figure 3 includes all of the data presented in Figure 2? Perhaps these figures could be combined for clarity. Figure 3 also requires expansion of the figure legend and description of acronyms.

• It would be more impactful if you included mention of the cytotoxicity of known toxic *MraY* inhibitors and less toxic derivatives from the literature when you mention your data from the HeLa2 cytotoxicity assays. This would provide much needed context for the reader.

• At the end of the "Design of stable analogues and evaluating their biological activity" section, the statement "These results suggested that these novel *MraY* inhibitors exhibit no cross-resistance to known drugs because of their novel mode of action" should be toned down significantly.

• In the section "Antibacterial properties of 2 and evaluation of activity in vivo" and Fig 4, it would be nice to contrast this data w/ the original muraymycin D2, in all panels.

• In this same section, statements such as "reduced bacterial burden by 99.99%" and "reducing bacterial burden by 99.999" are used. Does the data (presumably as depicted in Fig4d?) warrant these very precise values, given the variability seen in the assay?

• In the discussion, it is stated that "we showed that these *MraY*-analogue complexes (2 and 3) reveal distinct interaction patterns than natural *MraY* inhibitors, suggesting that these analogues are new types of *MraY* inhibitors with different pharmacologic profiles". This is quite a bold statement, and I believe, given the conservation of pose in the HS1 and Uridine regions, not true. This statement should be removed or significantly toned down.

• Likewise, the following should be toned down as well: "This simple compound architecture generates a unique interaction pattern with *MraY*, which is likely the basis of its novel pharmacological profiles."

Minor Comments:

• Please add line numbers in the revision, as it helps reviewers significantly.

• In Extended Data Figure 4c/h, the two FSC curves should be merged for clarity. Lines across at 0.5 can also be added and resolutions for map-model (@0.5).

• In Extended Data Figure 4 please show the mask used for the local refinement.

REVIEWER COMMENTS

For Reviewer #1:

1. Though the technique worked in this instance, this reviewer questions the broad applicability of this technique to other natural product molecules. The authors were advantaged in this study by the the fact that:
 1. all their core compounds were closely related structures.
 2. X-ray structures of all the bound lead molecules were available prior to initiating synthesis.
 3. All their compounds bound to the same target at the same binding site.
 4. The common structural core of the starting molecules bound to a well-defined pocket.
 5. The bioassay for improved function only relied on cellular permeability in a microbe.
 6. The core substructures chosen for the analog synthesis were themselves complex molecules but were readily available.

These are all advantages that most natural products would not have going in to such a study.

Response: We applied our strategy to a class of tubulin-binding natural products (epothilone B, paclitaxel, and vinblastine) to demonstrate the generality of our strategy, which is requested by the reviewer. We chose the class because, unlike the *MraY* inhibitory antibacterial natural products, it has cytotoxic activity against human cancer cells, its target protein is not an enzyme, its structures are very different within the class, and it binds to tubulin in a similar way but by a different subsequent mechanism. The core aldehydes of these natural products were synthesized by introducing formyl groups at completely different positions in each natural product (6 cores, see below, Supplementary Fig. 18). These were then used to synthesize a library consisting of 588 (= 6 x 98) hydrazone analogues by a build-up library synthesis strategy. Although the cytotoxic activity of the cores themselves was slightly reduced in all but one, we were able to identify several that were more active than the core when linked to fragments. For some of analogue libraries, the tubulin polymerization inhibitory or promoting activities were evaluated. We have completed these additional experiments, which include synthesis of 6 core aldehydes of natural products, a build-up library synthesis consisting of 588 analogues, and their biological evaluation only in 3 months. We hope these our efforts demonstrate the generality of our strategy. Details are discussed within the Supplementary information, and the description is transcribed below.

3. Application build-up library synthesis for tubulin-binding natural products

Tubulin-binding natural products are well-known compounds and used as antitumor drugs. This series includes epothilone B, paclitaxel, and vinblastine, which have very complex structures and exhibit very strong cytotoxicity, so drug discovery based on tubulin binders have been studied well. However, the analogue synthesis of these natural products are tough tasks because of their complicated and long synthetic route. Therefore, we thought that these natural products are suitable for application for our strategy as feasibility study. We designed six core aldehydes of epothilone B, paclitaxel, and vinblastine, according to previous semi-synthetic methods (Supplementary Fig. 18, Scheme S7-9). This allows us to synthesis libraries with different positions of natural product skeleton converted. In the case of epothilone B, the 7-hydroxyl group was modified, which was obtained directly from epothilone B^{S1}.

Additionally, a nitrogen atom in the aziridine analogue was modified by 3-formylbenzyl group, inspired by Nicolaou's reports^{S2-S3}. In the case of paclitaxel, an amino group in the 3-amino-2-hydroxy-3-phenylpropionyl moiety and a hydroxyl group in the taxane scaffold^{S4-S5}. In the case of vinblastine, 5-formyl analogue^{S6-S7} which could be obtained directly formylation of vinblastine, and 7-acyl analogue, which could be obtained from 2-steps transformations, were selected.

The library synthesis with these core aldehydes was conducted similar to synthesis of the library of MraY inhibitors. Two microliters of 10 mM DMSO solution of each aldehyde and two microliters of 10 mM DMSO solution of hydrazine (98 compounds) were applied to 96-well microplate, and the mixture was diluted with 16 μ L of DMSO to 1 mM solution. The microplate including reaction solutions were shaken for 30 min. The reaction mixtures were concentrated *in vacuo*, and the residues were dissolved with 20 μ L of DMSO to afford 1 mM DMSO solutions of hydrazone.

Firstly, tubulin polymerization assay was conducted with assay kit which detects microtubule polymerization by increasing fluorescence intensity (Supplementary Fig. 19a). Epothilone B and paclitaxel stabilize microtubule, resulting in increasing fluorescence intensity compared to DMSO control (Supplementary Fig. 19c,d). Core aldehydes **epo-azi-CHO** and **pac-O-CHO** maintained the activity of the parent natural products, while **pac-N-CHO** was more active than paclitaxel. However, **epo-O-CHO** exhibited a significant decrease in activity. Vinblastine inhibits microtubule polymerization, resulting in decreasing fluorescence intensity (Supplementary Fig. 19e). Both **vin-ind-CHO** and **vin-O-CHO** maintained the activity of vinblastine. From these results, we next evaluated the microtubule stabilizing/destabilizing activity of epo-azi, pac-N, and vin-O libraries (Supplementary Fig. 20). Because different classes of natural products have different effects on polymerization (Supplementary Fig. 20a-c), we compared the activity of the hydrazone analogues in terms of fluorescence intensity at the time when the difference is most apparent for each. In the epo-azi library, most of the analogues exhibited reduced activity, but there were a few that exhibited activity equal to or better than **epo-azi-CHO** and comparable to epothilone B (Supplementary Fig. 20d). In the pac-N library, many analogues exhibited improved activity over paclitaxel, with some exhibiting activity comparable to **pac-N-CHO** (Supplementary Fig. 20e). In the vin-O library, a few of analogues exhibited improved activity over vinblastine and **vin-O-CHO** (Supplementary Fig. 20f).

Next, cytotoxicity of these libraries against HCT-116 cells was evaluated (Supplementary Fig. 21). Unlike the results of the microtubule polymerization assay, the cell growth inhibitory activities of core aldehydes were lower than that of the parent natural products. Core aldehyde **epo-azi-CHO**, however, maintained most of the activity of the epothilone B (Supplementary Fig. 21a). Based on these results, the cell growth inhibitory activity of the libraries was evaluated at two concentrations (Supplementary Fig. 22). With the hydrazones synthesized in this study, we were unable to obtain analogues with activity exceeding that of the natural product, but we did find analogues with activity equivalent to that of natural product in epo-azi library (Supplementary Fig. 22a) and improved activity over the core aldehyde in pac-N and vin-O library (Supplementary Fig. 22c,f).

The results from the evaluation of tubulin-binding natural products are a clear indication that there does exist a discrepancy between protein-based and cell-based activities in natural product drug discovery. However, our strategy allows us to evaluate protein-based and cell-based activities simultaneously, so we can obtain a lot of information while taking cell-based activity into account. Especially for compounds with high cytotoxicity, such as tubulin-

binding natural products, conjugation with various molecules will be considered to achieve selective toxicity. In many cases, the introduction of a linker reduces the activity, but our strategy has the potential to be applied to optimize the linkers available for conjugation. We have completed these experiments, which include synthesis of 6 core aldehydes of natural products, a build-up library synthesis consisting of 588 analogues, and their biological evaluation only 3 months. These efforts demonstrate the generality of our strategy.

Supplementary Fig. 18. Structures of tubulin-binding natural products (epothilone B, paclitaxel, vinblastine) and designed core aldehydes.

Core aldehydes were named according to the first three letters of natural products and the substitution pattern of the formyl groups (3-formylbenzoyl, 3-formylbenzyl, formyl).

Supplementary Fig. 19. Tubulin polymerization assay with tubulin polymerization kit (Cytoskeleton, Inc.).

correspondence between the marks on the graphs and the concentrations of compounds is shown. **c)** These are graphs of the change in fluorescence intensity of epothilone B and its core aldehydes. Epothilone B stabilizes microtubules, resulting in a rapid increase in fluorescence intensity. **d)** These are graphs of the change in fluorescence intensity of paclitaxel and its core aldehydes. Paclitaxel stabilizes microtubules, resulting in a rapid increase in fluorescence intensity. However, at high concentrations, the increase in fluorescence intensity is small, probably due to low solubility. **e)** These are graphs of the change in fluorescence intensity of vinblastine and its core aldehydes. Vinblastine destabilizes microtubules, resulting in no increase in fluorescence intensity.

Supplementary Fig. 20. The results of tubulin polymerization assays of the hydrazone library.

Supplementary Fig. 21. The results for cell growth inhibitory assay of original natural products and core aldehydes (HCT-116, 72 h, WST-8).

Supplementary Fig. 22. The results for cell growth inhibitory assay (HCT-116 cells, 72 h, WST-8) of the hydrazone library.

2. In addition, due to the ligation strategy the authors used, they then had to synthesize more stable derivatives using a different chemical method. This raises a question of whether or not the plate-based method they used would provide sufficient sample material for more than one assay.

Response: In our strategy, hydrazone library is intended to use just as the first screening for evaluations of biological activities. After identification of the screening hits (compounds that exhibited the desired activity), scale-up synthesis of the hit compounds can be performed for further evaluation. In this study, we were concerned about hydrolysis of hydrazone bond when oriented toward in vivo evaluations, so we chose to link the fragments with a chemically more stable amide bond in scale-up synthesis. Of course, we believe it is also possible to perform higher-order evaluations with the hydrazones intact.

3. Also, the authors quote ~80% yields optimally using their method. 20% contamination with building blocks could result effects that alter assay results (i.e. core structures could still bind to target proteins and reduce efficacy of the fully articulated analog).

Response: While the reviewer's concerns are certainly valid, this study showed that the *MraY* inhibitory and antimicrobial activities of the core aldehyde and accessory fragments themselves are greatly reduced compared to the original natural product (Supplementally Fig 9 and see below). Therefore, if the activity of the mixture corresponding to each well is increased, it could be a hit compound. Even if the purity is 50%, if the activity is increased, it means that the activity of the linked derivatives is increased. On the other hand, it could be the case that the activity value of the core is not different from the original natural product. In such cases, it is reasonable to assume that if the activity improves, it is from the linked derivative, although care should be taken in interpretation.

Supplementary Fig. 9. *MraY* inhibitory activity of core aldehydes.

Conditions: Reactions were carried out in a 384-well microplate. A solution containing 10 μM dansylated -UDP-MurNAc-pentapeptide and 50 μM undecaprenyl phosphate (C_{55} -P) in 20 μL of an assay buffer [50 mM Tris-HCl (pH 7.6), 50 mM KCl, 25 mM $MgCl_2$, 0.2% Triton X-100 and 8% glycerol] was prepared. The reactions were initiated by the addition of *S. aureus* Mray enzyme (5 μL , 11 $\mu g/mL$). After 3 h of incubation at room temperature, the formation of dansylated lipid I was monitored by fluorescence enhancement (excitation at 355 nm, emission at 535 nm). The mixtures contained 2% DMSO in order to increase the solubility of the compounds (concentrations; 0.0001, 0.001, 0.01, 0.1, 1, 10, 100 μM).

4. Finally, the structure that provided the best efficacy was very close structurally to the initial molecule tunicamycin (having the sugar units exchanged). Though a fine result, not particularly non-intuitive as far as analog design by other means.

Response: The main thrust of this paper is that we have developed a methodology for how to rapidly search for analogues with superior biological activity. We did not want to find analogues with different structures in this study. If we want to develop analogues with different structures, we can set up core compounds with boldly different structures.

For Reviewer #2:

5. Incorrect grammar usage makes the manuscript difficult to follow. The structure of sentences and paragraphs further detracts from the significance of the study. For example with the latter, a significant portion of the introduction is dedicated to why the authors employed a so-called build-up strategy (which, by the way, is essentially a convergent synthesis strategy) to make natural product analogues. This information could be dramatically shortened to perhaps even one sentence. One potential way to restructure the introduction is first to introduce *MraY*, then describe the known natural product inhibitors (including grouping based on inhibition patterns, potency, and other pertinent information), and finally how a convergent synthesis approach was developed to make simplified analogues that consist of a core component (binding to the “uridine” binding site) and a peripheral moiety (binding to one of the additional hot spots) to make a focused library.

Response: We appreciate the suggestion. Grammar was carefully corrected, and the order of the descriptions in the paper introduction was dramatically changed. As pointed out by the reviewer, we described antibacterial drug development and *MraY*, and then described *MraY* inhibitory natural products. After that, we described the matters revealed in the previous studies of *MraY* inhibitors and difficulties in logical design based on them, and then explained the build-up library synthesis strategy. As a results, the Introduction was changed as below.

Antimicrobial resistance (AMR) bacteria are spreading worldwide, and there is an urgent need to develop new antibacterial agents that are effective against resistant bacteria.^{1,2} In order to develop antibacterial agents that are effective against drug-resistant bacteria such as methicillin-resistant *Staphylococcus aureus* (MRSA), vancomycin-resistant enterococci (VRE), it is important to develop a new class of antibacterial agents that do not interfere with existing resistance mechanisms. Phospho-*N*-acetylmuramoyl-pentapeptide-transferase (*MraY*) is a bacterial transmembrane enzyme, which is responsible for the formation of lipid I during peptidoglycan biosynthesis (Supplementary Fig. 1).^{3,4} Since *MraY* is universally present in bacteria and an essential enzyme for their survival⁵, it is expected to be a novel and promising target in antibacterial drug discovery. Nucleoside antibiotics⁶ that include tunicamycins⁷, muraymycins⁸, mureidomycins⁹, and capuramycin¹⁰ are known as *MraY* inhibitors, which show antibacterial activity against drug-resistant bacteria (Fig. 1a). These natural products have uridine moiety as a

common substructure, but the other moieties are structurally diverse, and each has a characteristic antibacterial spectrum. MraY inhibitory natural products have been extensively studied, and their co-crystal complex structures bound to MraY have been solved and the common uridine moiety binds to the uridine pocket, while other moieties interact with various hot spots (HSs) of MraY (Fig. 1a, b).¹¹⁻¹³ These studies provide a foundation for structure-based approaches to design improved MraY inhibitors. While structural modification of a single natural product may lead to a dead end, since multiple natural products inhibit the same MraY enzyme, we can expect that comprehensive structural modification of these natural products will increase the possibility of developing better inhibitors. However, it is still difficult to rational design of inhibitors because MraY is a dynamically conformation-changing enzyme. In its apo form, no clear interaction pockets with an inhibitor are found. However, MraY undergoes induced fits upon inhibitor binding. Consideration of membrane permeability is also critical because the catalytic site of MraY resides on the cytoplasmic side of the cytoplasmic membrane, thus MraY inhibitors must be able to penetrate the cytoplasmic membrane.¹¹ Simultaneous optimization of MraY inhibition and bacterial accumulation is essential to obtain whole-cell activity; however, balancing the requisite physicochemical properties needed for accumulation while retaining molecular features defined by the structure-activity relationship (SAR) to maintain binding to MraY is especially challenging given the complex and polar nature of these antibiotics. Furthermore, synthesizing a sufficient number of analogues for structural optimization is not easy due to their complex structure requiring multi-step synthesis.

Even with a single natural product, the synthesis of natural product analogues having a large and complicated chemical structure sometimes requires multiple steps accompanied by complicated purification and structure determination processes thereby tremendously high costs, and how to synthesize a library of natural product analogues as quickly and comprehensively as possible is one of the propositions in medicinal chemistry based on natural products. So, we developed the platform simplifying comprehensive analogue synthesis of a series of natural products, which accelerates the structural optimization of MraY inhibitory natural products. We focused on *in situ* screening, which has recently come to be used. In this method a library of compounds is synthesized on an assay plate and biological activity is evaluated without purification.¹⁴⁻¹⁷ Not only can a large number of analogues be rapidly synthesized by this method, but also the amount of compounds required for biological activity evaluation should be small. Therefore, we considered this approach to be one of the solutions that would allow an effective and comprehensive optimization of multiple natural products. Our strategy is to first divide the chemical structure of natural products into two fragments; one is a core fragment, which is expected to play a role in a key contributor upon binding to the target, and the other is an accessory fragment that is expected to further modulate binding affinity to the target, selectivity to off-target(s), and most importantly, the disposition properties (Fig. 1a). These two fragments are ligated to construct a library of natural products prior to biological evaluation, which we call a “build-up library”. Many studies of *in situ* screening often retain by-products derived from reagents such as condensing agents in the reaction mixture, and therefore, are limited to biochemical assays.¹⁴ In this study, the core fragment and a library of the accessory fragments are ligated by a reaction, which proceeds with high chemoselectivity and near quantitative yield without any contaminating reagents or by-products. Our approach avoids lengthy multi-step synthesis, purification, and characterization of each compound and enables direct biological evaluation in an enzymatic and a cell-based assay. In this study, we applied this strategy to the structural optimization of MraY

inhibitory natural products in order to develop new antibacterial drug leads tackling drug-resistant infections. Using the 7 cores (four classes) and 98 accessory fragments, the build-up library composed of a large number of analogues is prepared. Several analogues exhibiting strong *MraY* inhibitory and antibacterial activity are identified, and further optimization of these analogues leads to the identification of analogue 2, which is effective against drug-resistant strains and in mouse infection models. We also applied our "build-up library" synthesis strategy to a class of tubulin-binding natural products, including epothilone, paclitaxel, and vinblastine, demonstrating the versatility of our strategy.

6. On page 9, the authors describe dissolving the crude reaction mixture into DMSO to make a reported 5 mM stock that assumes quantitative conversion of the reagents. However, it is noted that while many were obtained at 80% or higher, some did not have great yields or proceed at all. Was the conversion to the product considered when reporting concentration used in Figure 2 (for example TUN-type library at 100 nM) and Figure 3 for MIC values? How can one product be compared to another if, for example, one reaction gives quantitative conversion while the other 10%? Also with regard to the activity, it was not clear whether the separate fragments (core or peripheral moiety) had activity alone or together without covalent coupling. Finally, it would be nice to have information regarding the IC₅₀, MIC, and spectrum of the natural products as comparators. If they were not used in this study, then citing the appropriate reference and providing the values would suffice.

Response: The conversion rates of the products are not reflected in the bioactivity values. Since the library synthesis and biological activity evaluations by hydrazone formation in this study is positioned as a screening, the primary goal is to find highly active analogues. Therefore, as the reviewer pointed out, it is difficult to obtain a structure-activity relationship in a strict sense, since we cannot discuss the strength of activity between an analogue with 100% conversion and an analogue with 10% conversion. However, in this study, since the activity of the cores is low and the conversion rate is high in most cases, we believe it is possible to obtain a rough structure-activity relationship. Therefore, we have added the following sentence in the section "Synthesis and biological evaluation of build-up library" regarding the activity values of the core aldehydes and the fact that conversion rates are not taken into account.

"In subsequent evaluations of the biological activity of the library, the evaluation was performed at concentrations assuming 100% conversion. We confirmed that the *MraY* inhibitory activity of aldehyde cores was 100~1000 times lower than that of the original natural products (Supplementary Fig. 9).^{34,32-35} However, in the case of capuramycin aldehyde core, the inhibitory activity did not decrease significantly as reported previously.^{20,34} Based on these results, we determined the evaluated concentration of the library so that the *MraY* inhibition of the aldehyde core would be less than 20%. Also, the *MraY* inhibition of hydrazine fragments were up to 58%, even at 200 μ M."

7. On page 12, the cytotoxicity against HepG2 cells is stated at 38-66 μ M. Is this IC₅₀? How does the therapeutic

index (antibiotic vs toxicity activity) compare when considering the parent natural product? For example, it is well known that tunicamycin is toxic and therefore would be expected to have both antibacterial and cytotoxic activities. Does the ratio increase or decrease with the analogues? The noted cytotoxicity makes the mouse model study concerning. Did the dose that was used have any effect on the mice? Is the maximal tolerated dose known?

Response: The “IC₅₀” was omitted from the cytotoxicity values, so it has been corrected (IC₅₀ = 38-66 μM). Although cytotoxicity evaluation of parent natural product has not been performed, muraymycin does not exhibit cytotoxicity (Tanino, T. *et al. J. Med. Chem.* **54**, 8421-8439 (2011)). The therapeutic index of the analogues may be narrower, but it has improved antibacterial activity, and no significant toxicity was observed in the mouse model. For the mouse experiment, the following sentence were added “No significant weight loss or other toxicity was observed within the dosing concentrations.”. In addition, no maximum tolerated test was conducted. Although a reduction in cytotoxicity is necessary for clinical use as an antibacterial agent, this could be achieved by conversion of the lipid moiety, as described below.

Tunicamycin is known to be toxic due to inhibition of GPT in humans, and some studies suggest that the long-chain acyl group is responsible for its toxicity (Kurosu, M. *et al. Angew. Chem. Int. Ed.* **61**, e202203225 (2022), Carpenter, E. P. *et al. Cell* **175**, 1045-1058 (2018), and our unpublished data). We have also shown that the structure of the acyl group contributes to toxicity in our studies of caprazamycin derivatives (Ichikawa, S. *et al. Org. Biomol. Chem.* **13**, 7720-7735 (2015)). Therefore, in this study, we used both palmitoyl group, which is of concern for toxicity, and the isostearoyl group, which is of less concern. Although the palmitoyl group was selected for subsequent studies because of its good antibacterial activity, we believe that toxicity can be avoided by changing the long-chain acyl group moiety. In such cases, our strategy allows for rapid screening once the fragment is synthesized.

8. The statement that the analogues have “different” or “novel” pharmacological profiles likely needs clarification. What is meant by pharmacological profile, and what are the analogues being compared to? If, for example, the authors are attempting to say that the analogues, specifically analogue 2, causes the formation of abnormal membrane-like structures, has the parent natural product (for example muraymycin D1) been examined in the same assay for comparison? Perhaps instead of broadly stating differences in pharmacological profiles, specifically note what property is unique with respect to manifestation of its antibacterial activity. It is worth pointing out that other natural products (polymixins, aminoglycoside, etc) are known to disrupt membrane structure and, for aminoglycosides, bind the ribosome thus having a dual mechanism of action in the manner of what is being proposed for the analogues in this manuscript.

Response: The use of the word “profile” was ambiguous, so we have either added a clarifying comment or replaced it with another word, as follows.

“Structures of the MraY-analogue complexes reveal distinct interaction patterns, suggesting that these analogues represent new types of MraY inhibitors with unique pharmacological profiles.”

→ Structures of the MraY-analogue complexes reveal distinct interaction patterns, suggesting that these analogues represent new types of MraY inhibitors with unique binding mode.

“we showed that these MraY-analogue complexes (2 and 3) reveal distinct interaction patterns than natural MraY inhibitors, suggesting that these analogues are new types of MraY inhibitors with different pharmacological profiles.”

→ we showed that these MraY-analogue complexes (2 and 3) share a common interaction pattern with natural MraY inhibitors in uridine and aminoribose moiety, but show different interaction patterns at other sites, suggesting that these analogues are new types of MraY inhibitors with different interaction profiles.

“This simple compound architecture generates a unique interaction pattern with MraY, which is likely the basis of its novel pharmacological profiles.”

→ This simple compound architecture generates a unique interaction pattern with MraY.

Similar experiments were not performed with the parent natural products (muraymycin D2) because it does not exhibit antibacterial activity. Therefore, it is difficult to directly compare the action of the analogues on bacterial membranes. However, the dual action of other natural products was mentioned in the text as follows.

“The TEM images suggested that 2 may cause membrane disruption in addition to inhibiting peptidoglycan biosynthesis. This dual action was also observed with polymyxins⁵² and aminoglycosides^{53,54} and may be one of its promising properties as an antimicrobial material.”

For Reviewer #3:

Major Comments:

9. Figure one has most of the components to be an excellent figure, yet its presentation is a bit confusing. I have a number of suggestions to address this:

- Figure 1 realistically has many panels, yet is split into A and B. This can lead to confusion for the reader and in regard to what is what when comparing the figure to the description below. As well, the numbering from A is in the figure legend but not in the figure itself, that would be a great addition. As well, I would recommend the whole figure be converted to at least A-E for clarity.
- Regarding the order of the content, the figure goes from a schematic overview, then back to the data that supported the rationale for the workflow above, then back to zooming into a section of the workflow. I think it

would be most logical to start the figure with the first two panels of B (tunicamycin V, muraymycin D2...) to show the rationale, then proceed to the current A overview, followed by the lower two panels of D.

- The structural overlay could use some improvement, particularly for clarity at its current size.
- Somehow it should be added to this figure that this workflow w/ the bio-compatible chemistry allows for the antibacterial assays (diagrammatically, as the font “direct evaluation of enzymatic and cell-based assay” very much under sells this point). This is a key advance made in this work and should thus be presented as such.

Response: We have changed the configuration of Figure 1 as pointed out by reviewer #2. In line with the structure of the Introduction, the MraY inhibitory natural products are described in a) and the complex structures bound to MraY are described in b). Then, to match the follow of the build-up library synthesis strategy that was developed to rapidly optimize the structure of the MraY inhibitory natural products, the strategy was outlined as c). The chemical structures to be used in the hydrazone library were then included in d) and e). One of the strengths of this strategy is that it allows direct biological activity evaluation, which is also highlighted in the figure. The modified figure is shown below.

Fig. 1. a) Structures of MraY inhibitory natural products have common uridine moiety binding to uridine pocket (red) and other motif with various binding mode in MraY. HS1–4 represent binding hot spot (see details in reference 13, magenta; uridine pocket, green; HS1, purple; HS2, salmon; HS3, cyan; HS4, light brown; HS5,

brown; HS6). **b)** Overlay of complex structures of these antibiotics bound to MraY. Each carbon color represents each antibiotic in panel a (salmon; tunicamycin, green; muraymycin D2, cyan; 3'-hydroxy-mureidomycin A, orange; capuramycin). **c)** Overview of a comprehensive *in situ* evaluation of the build-up library. 1) Natural products are divided into the core and accessory. 2) Reaction site is introduced into both pairs. The reaction ideally proceeds quantitatively and selectively without toxic reagents and by-products. 3) The core and accessory fragments are ligated on the assay plates. The resulting library is directly evaluated by enzymatic and cell-based assays. 4) A comprehensive SAR is obtained and hit analogues are identified. **d)** Core fragments are the core substructures containing uridine from these antibiotics and formyl group attached to them. Each antibiotic type is represented by three capital letters (tunicamycin; **TUN**, muraymycin; **MRY**, 3'-hydroxy-mureidomycin; **MRD**, capuramycin; **CAP**) and an additional letter (*para* or *meta* substituted formyl group; **pCHO** or **mCHO**). Aldehydes are ligated with hydrazine (named **BZXX**, **PAXX**, **ACXX**, **AAXX**, or **LAXX**; XX is a number). **e)** The name of the obtained hydrazone should be indicated with the aldehyde name in front and the hydrazine name behind (e.g. **TUNp-BZXX**).

10. Some comment on MraY similarities to human GPT (GlcNAc phosphotransferase) is required in the introduction. Given tunicamycin can inhibit mammalian N-linked glycosylation, this is important to discuss. Does the workflow and methodology of this work provide any insight into potential toxicities? Have any steps been taken to design a library that will produce hits that avoid toxicity in humans?

- Toxicity not seen for selected compound in HeLa2 cytotoxicity assays and mouse studies – was this by chance or design?

Response: Toxicity avoidance was not taken into account in this study. The three parent natural products, including muraymycin, which was the focus of this study, do not exhibit toxicity in the literature. Tunicamycin is known to exhibit cytotoxicity, so toxicity should be taken into account in future studies focusing on tunicamycin. The toxicity of analogues **1-8** to HepG2 cells is probably due to the palmitoyl group and the fact the entire molecules is positively charged. Therefore, we believe that the palmitoyl group should be changed in the future. We have added the following comments on this in the Discussion section as follows.

“The selective inhibition of MraY vs. human GPT is a key issue in the development of antibacterial agents. GPT, which belongs to the same superfamily as MraY, is an enzyme that transfers GlcNAc phosphate to dolichol phosphate in the glycan biosynthesis pathway of glycoproteins, and inhibition of human GPT is thought to induce endoplasmic reticulum stress, leading to cell apoptosis.⁵⁵⁻⁵⁷ Tunicamycin is known to inhibit GPT and is known to be cytotoxic.⁵⁸⁻⁶¹ Therefore, in order to develop MraY inhibitors as antibacterial agents, attention should also be paid to cytotoxicity.”

11. Figure 2 legend is lacking a majority of detail to fully comprehend the figure. Please add description of all

acronyms used, description of the various panels (A-E).

Response: In response to the two later points, Figure 2 and Figure 3 were merged into a new Figure 2. Legends have also been added as appropriate. The merged new Figure 2 is described below.

a) General structures of hydrazine fragment moiety

b) TUNp-library

c) TUNm-library

d) MRYP-library

e) MRYm-library

f) MRDp-library

g) MRDm-library

h) CAP-library

Fig 2. a) General structures of hydrazine fragment moiety are shown. Each hydrazine class are represented by different colors (green; BZ, cyan; PA, gray; AC, yellow; AA, orange; LA). **b-h)** MraY inhibitory activity and antibacterial activity of the hydrazone library. The bar graphs show the relative enzyme activity with DMSO as 100% when the compound is treated at given concentration (b,c; TUN at 100 nM, d,e; MRY at 10 nM, f,g; MRD at 100 nM, h; CAP at 1000 nM. n = 1), with the lower bars indicating higher inhibitory activity. Antibacterial activity was evaluated by the minimum inhibitory concentration (MIC) at three points, 0.5, 5, and 50 μ M (n = 1), and shown as a heat map with the compound on the horizontal axis and the bacterium on the vertical axis. The redder the heat map, the lower the MIC value, i.e., the higher the antibacterial activity. Data of control compounds (van: vancomycin, col: colistin) are shown left column in each panel. *E. faecium* ATCC 35667, *S. aureus* ATCC 29213, *P. aeruginosa* ATCC 27853, *K. pneumoniae* ATCC 13883, *E. cloacae* ATCC 13047, *A. baumannii* ATCC 19606, *E. coli* ATCC 25922.

12. In Figure 2, vastly different concentrations are used for each “class” of compounds. Were these all tested at 10, 100, 1000 nM? If so, this data should, in the very least, be included in the Extended Data.

Response: First, each class of libraries was evaluated only at their respective concentrations. Based on the activity

of the original natural product, the evaluation was performed at concentrations near the IC₅₀ of the natural product, where the difference in activity becomes clear. In the case of mureidomycin, the activity decreases significantly when the urea moiety is removed, so in order to identify analogues with high inhibitory activity even without the urea moiety, evaluations were conducted at concentrations 10 times higher than the IC₅₀ of the natural product. The following comments were added before the library evaluation in the text.

“In subsequent evaluations of the biological activity of the library, the evaluation was performed at concentrations assuming 100% conversion. We confirmed that *MraY* inhibitory activity of aldehyde cores were 100~1000 times lower than that of the original natural products (Supplementary Fig. 9).^{34,32-35} However, in the case of capuramycin aldehyde core, the inhibitory activity did not decrease significantly as reported previously.^{20,34} Based on these results, we determined the evaluated concentration of the library so that the *MraY* inhibition of the aldehyde core would be less than 20%. Also, the *MraY* inhibition of hydrazine fragments were up to 58%, even at 200 μM .”

13. Figure 3 includes all of the data presented in Figure 2? Perhaps these figures could be combined for clarity. Figure 3 also requires expansion of the figure legend and description of acronyms.

Response: Figure 2 and Figure 3 were merged into a new Figure 2 and legends were added as appropriate. The results of each core were divided into separate panels. A new Figure 2 is pasted in the two previous remarks.

14. It would be more impactful if you included mention of the cytotoxicity of known toxic *MraY* inhibitors and less toxic derivatives from the literature when you mention your data from the HelG2 cytotoxicity assays. This would provide much needed context for the reader.

Response: The following comment was added “These values did not indicate strong toxicity compared to known *MraY* inhibitors.^{42,44,45}”. Muraymycin D2 and its lapidated analogues are not cytotoxic, and the toxic caprazamycin analogues have IC₅₀ values of approximately two orders of magnitude μM (Tanino, T. *et al. J. Med. Chem.* **54**, 8421-8439 (2011). & Nakaya, T. *et al. Org. Biomol. Chem.* **13**, 7720-7735 (2015).). Tunicamycin, which is relatively cytotoxic, has IC₅₀ values ranging from sub- μM to double-digit μM , depending on the cell type (Mitachi, K. *et al. Angew. Chem. Int. Ed.* **61**, e202203225 (2022).).

15. At the end of the “Design of stable analogues and evaluating their biological activity” section, the statement “These results suggested that these novel *MraY* inhibitors exhibit no cross-resistance to known drugs because of their novel mode of action” should be toned down significantly.

Response: The phrase “to known drugs” was toned down to “to known drugs evaluated in this study”; “These

results suggested that these novel *MraY* inhibitors are likely not cross-resistance to known drugs evaluated in this study because of their novel mode of action.”.

16. In the section “Antibacterial properties of 2 and evaluation of activity in vivo” and Fig 4, it would be nice to contrast this data w/ the original muraymycin D2, in all panels.

Response: The original muraymycin D2 itself shows no antibacterial activity, not even against Gram-positive bacteria. It is also difficult to synthesize and evaluate again the analogues reported in the past (MIC against *S. aureus* >64 µg/mL Tanino, T. *et al. J. Med. Chem.* **54**, 8421-8439 (2011).). Although it is important to compare it with the original natural product, we believe that it is more important to compare its pharmacological effect with known antibacterial drugs.

17. In this same section, statements such as “reduced bacterial burden by 99.99%” and “reducing bacterial burden by 99.999” are used. Does the data (presumably as depicted in Fig4d?) warrant these very precise values, given the variability seen in the assay?

Response: The expression “99.99%” was changed to “4log₁₀ units” to avoid giving the impression of being too precise.

18. In the discussion, it is stated that “we showed that these *MraY*-analogue complexes (2 and 3) reveal distinct interaction patterns than natural *MraY* inhibitors, suggesting that these analogues are new types of *MraY* inhibitors with different pharmacologic profiles”. This is quite a bold statement, and I believe, given the conservation of pose in the HS1 and Uridine regions, not true. This statement should be removed or significantly toned down.

Response: The text was changed as described below “we showed that these *MraY*-analogue complexes (2 and 3) share a common interaction pattern with natural *MraY* inhibitors in uridine and aminoribose moiety, but show different interaction patterns at other sites, suggesting that these analogues are new types of *MraY* inhibitors with different interaction profiles. “.

19. Likewise, the following should be toned down as well: “This simple compound architecture generates a unique interaction pattern with *MraY*, which is likely the basis of its novel pharmacological profiles.”

Response: The following sentence was deleted “which is likely the basis of its novel pharmacological profiles”.

20. Minor Comments:

- Please add line numbers in the revision, as it helps reviewers significantly.
- In Extended Data Figure 4c/h, the two FSC curves should be merged for clarity. Lines across at 0.5 can also be added and resolutions for map-model (@0.5).
- In Extended Data Figure 4 please show the mask used for the local refinement.

Response: These were corrected. The new version of Extended Data Figure 4 has been moved to Supplementary Information as Supplementary Fig. 16 as shown below.

Supplementary Fig. 16. Cryo-EM data processing.

a) representative micrograph and 2D classification for Mray-NB7-2.

b) local resolution estimation for Mray-NB7-2.

c) phenix reported Fourier shell correlations, and particle angular distribution for the final map of Mray-NB7-2.

d) cryo-EM density corresponding to Mray TM1-10 for Mray-NB7-2 (map threshold = 0.25).

- e) processing workflow for **MraY-NB7-2**.
- f) representative micrograph and 2D classification for **MraY-NB7-3**.
- g) local resolution estimation for **MraY-NB7-3**.
- h) phenix reported Fourier shell correlations, and particle angular distribution for the final map of **MraY-NB7-3**.
- i) cryo-EM density corresponding to **MraY TM1-10** for **MraY-NB7-3** (map threshold = 0.25).
- j) processing workflow for **MraY-NB7-3**.

REVIEWERS' COMMENTS

Reviewer #1 (Remarks to the Author):

The authors have done a good job of addressing my initial concerns on the broad applicability of their technique. In response to my query about the level of impurities, the authors make a reasonable rebuttal. They do, however, agree that in some cases "care should be taken in interpretation" of results where the activity of the core structure is not significantly different from the product. It would be good to add that statement to the text.

The overall revised manuscript is excellent. It appears the authors have addressed the concerns of all the reviewers. I am now satisfied that the manuscript is acceptable for publication in Nature Communications.

Reviewer #2 (Remarks to the Author):

The authors adequately addressed concerns from the first review. Additional data was also provided to support the "applicability" of their approach. I support publication.

Reviewer #3 (Remarks to the Author):

The authors have largely addressed my former comments.

REVIEWER COMMENTS

For Reviewer #1:

The authors have done a good job of addressing my initial concerns on the broad applicability of their technique. In response to my query about the level of impurities, the authors make a reasonable rebuttal. They do, however, agree that in some cases "care should be taken in interpretation" of results where the activity of the core structure is not significantly different from the product. It would be good to add that statement to the text.

The overall revised manuscript is excellent. It appears the authors have addressed the concerns of all the reviewers. I am now satisfied that the manuscript is acceptable for publication in Nature Communications.

Response: Thank you for your comments. We have added the following statement to the Discussion section regarding the caution needed when the biological activity of the core structure is not significantly different from that of the product:

In the case of tubulin-binding natural products, caution is needed in interpreting results when the conversion rate of the ligation reaction is low due to the relatively high biological activity of the core aldehyde. This is because high biological activity originating from the core aldehyde may be observed. It is important to verify the purity of candidate analogues that are judged to have high activity using techniques such as LC-MS.

For Reviewer #2:

The authors adequately addressed concerns from the first review. Additional data was also provided to support the "applicability" of their approach. I support publication.

Response: Thank you for reviewing our manuscript and your comments.

For Reviewer #3:

The authors have largely addressed my former comments.

Response: Thank you for reviewing our manuscript and your comments.

For Editor:

This time, due to the unknown origin of the cell line used in the cytotoxicity experiments, a re-experiment was conducted using a new cell line purchased from ATCC (HepG2: HB-8065, HCT116: CCL-247). Below are the results of cytotoxicity against HepG2 cells of MraY inhibitory natural products (Supp.Table2) and cytotoxicity against HCT116 cell lines of tubulin-binding natural products, each shown in the order of the old version (cell line of unknown origin) and the new version (cell lines from ATCC).

Regarding the cytotoxicity of the analogues of MraY inhibitory natural products, in the experiment with the newly purchased HepG2 cell line (HB8065 from ATCC), the IC₅₀ values were about one quarter to one half lower than the

old ones. Although it does not show significant toxicity in mouse tests, it still has moderate cytotoxicity, so improving selectivity is a future challenge. Regarding the results in the HCT116 cell line for tubulin-binding natural products, it was found that the sensitivity to tubulin-binding drugs was slightly lower in cells of unknown origin. On the other hand, in the newly purchased ATCC-derived cells (CCL-247), the activity of natural products and core aldehydes slightly improved, and accordingly, the activity in the library also improved (the lower the bar, the stronger the activity). Regardless of the differences in strength, the trend of cytotoxicity is the same, and it can indicate that this experiment has ensured reproducibility. As a result, we believe that these results with newly purchased HepG2 and HCT116 cell lines do not affect the content of this manuscript.

HepG2: MraY inhibitory natural products analogues

Old version

Supplementary Table 2. Cytotoxic activity of stable analogues 1-8 against HepG2 cells.

	Amide MRY				anilide MRY			
	1	2	3	4	5	6	7	8
IC ₅₀ (μM)	38	41	45	40	46	66	45	52

HepG2 cells (1x10⁴ cells) were cultured at 37 °C under 5% CO₂ in air in D-MEM (low-glucose) supplemented with 10% fetal bovine serum (FBS). After incubation for 24 h, solutions of test compounds in DMSO were added to the culture, and the 96-well plates were incubated for 24 h under above conditions. After that, the WST-8 cell proliferation assay was used to evaluate cell viability. Experimental details are shown in methods section of the main text.

New version (HepG2 ATCC HB-8065)

	amide MRY				anilide MRY			
	1	2	3	4	5	6	7	8
IC ₅₀ (μM)	12	16	18	19	22	16	25	23

Supplementary Table 2. Cytotoxic activity of stable analogues 1-8 against HepG2 cells.

Experimental details are shown in methods section of the main text. HepG2 cells (1x10⁵ cells/mL, 100 μL) were cultured at 37 °C under 5% CO₂ in air in E-MEM (with L-glutamine, phenol red, sodium pyruvate, non-essential amino acids and 1,500 mg/L sodium bicarbonate, wako) supplemented with 10% fetal bovine serum (FBS). After incubation for 24 h, solutions of test compounds in DMSO were added to the culture, and the 96-well plates were incubated for 24 h under above conditions. After that, the WST-8 cell proliferation assay was used to evaluate cell viability. This data was collected from three independent experiments (n=3).

Old version

Supplementary Fig. 21. The results for cell growth inhibitory assay of original natural products and core aldehydes (HCT-116, 72 h, WST-8).

a) Cell viability plots of epothilone B, **epo-O-CHO**, and **epo-azi-CHO**. b) Cell viability plots of paclitaxel, **pac-N-CHO**, and **pac-O-CHO**. c) Cell viability plots of vinblastine, **vin-ind-CHO**, and **vin-O-CHO**.

HCT-116 cells (1×10^4 cells) were cultured at 37 °C under 5% CO₂ in air in McCoy's 5A (1x) (Gibco™) supplemented with 10% fetal bovine serum (FBS). After incubation for 24 h, solutions of test compounds in DMSO were added to the culture, and the 96-well plates were incubated for 72 h under above conditions. After that, the WST-8 cell proliferation assay was used to evaluate cell viability.

Supplementary Fig. 22. The results for cell growth inhibitory assay (HCT-116, 72 h, WST-8) of the hydrazone library.

Supplementary Fig. 21. The results for cell growth inhibitory assay of original natural products and core aldehydes.

a) Cell viability plots of epothilone B, epo-O-CHO, and epo-azi-CHO. **b)** Cell viability plots of paclitaxel, pac-N-CHO, and pac-O-CHO. **c)** Cell viability plots of vinblastine, vin-ind-CHO, and vin-O-CHO.

HCT-116 cells (1×10^4 cells) were cultured at 37 °C under 5% CO₂ in air in McCoy's 5A (1x) (Gibco™) supplemented with 10% fetal bovine serum (FBS). After incubation for 24 h, solutions of test compounds in DMSO were added to the culture, and the 96-well plates were incubated for 72 h under above conditions. After that, the WST-8 cell proliferation assay was used to evaluate cell viability. These data were collected from three independent experiments (n=3).

Supplementary Fig. 22. The results for cell growth inhibitory assay (HCT-116, 72 h, WST-8) of the hydrazone library.

The horizontal axis indicates the hydrazone number (green; BZ-type, blue; PA-type, gray; AC-type, yellow; AA-type, orange; LA-type). The left side of each graph shows the results for the original natural products and their core aldehydes (black). The dashed lines show the viability when treated with the higher concentrations of original natural products. These data were collected from a single trial (n=1).